# A graded neonatal mouse model of necrotizing enterocolitis demonstrates that mild enterocolitis is sufficient to activate microglia and increase cerebral cytokine expression

Cuilee Sha[1,2,3], Trevor Van Brunt[1,2,4] Jacob Kudria[2,4], Donna Schmidt[2,5], Alisa Yurovsky[6], Jela Bandovic[7], Michael Giarrizzo[8], Joyce Lin[9], Styliani-Anna Tsirka[3], Agnieszka B. Bialkowska[8], Lonnie P. Wollmuth[2,4], Esther M. Speer[9], Helen Hsieh[2,5]*

1 Renaissance School of Medicine, Stony Brook, New York, United States of America, 2 Center for Nervous System Disorders, SUNY-Stony Brook, Stony Brook, New York, United States of America, 3 Department of Molecular and Cellular Pharmacology, SUNY-Stony Brook, Stony Brook, New York, United States of America, 4 Department of Neurobiology and Behavior, SUNY-Stony Brook, Stony Brook, New York, United States of America, 5 Department of Surgery, Division of Pediatric Surgery, Stony Brook Medicine, Stony Brook, New York, United States of America, 6 Department of Biomedical Informatics, SUNY-Stony Brook, Stony Brook, New York, United States of America, 7 Department of Pathology, Stony Brook Medicine, Stony Brook, New York, United States of America, 8 Department of Medicine, Division of Gastroenterology, Stony Brook Medicine, Stony Brook, New York, United States of America, 9 Department of Pediatrics, Division of Neonatology, Stony Brook Medicine, Stony Brook, New York, United States of America

* Helen.Hsieh@stonybrookmedicine.edu

## Abstract

Necrotizing enterocolitis (NEC) is an inflammatory gastrointestinal process that afflicts approximately 10% of preterm infants born in the United States each year, with a mortality rate of 30%. NEC severity is graded using Bell's classification system, from stage I mild NEC to stage III severe NEC. Over half of NEC survivors present with neurodevelopmental impairment during adolescence, a long-term complication that is poorly understood. Although multiple animal models exist, none prospectively controls for NEC severity. We bridge this knowledge gap by characterizing a graded murine model of NEC and studying its relationship with neuroinflammation across a range of NEC severities. Postnatal day 3 (P3) C57BL/6 mice were fed a formula containing different concentrations (0% control, 0.25%, 1%, 2%, and 3%) of dextran sodium sulfate (DSS). P3 mice were fed every 3 hours for 72 hours. We collected data on weight gain and behavior (activity, response, body color) during feeding. At the end of feeding, we collected tissues (intestine, liver, plasma, brain) for immuno-histochemistry, immunofluorescence, and cytokine and chemokine analysis. Throughout NEC induction, mice fed higher concentrations of DSS died sooner, lost weight faster, and became sick or lethargic earlier. Intestinal characteristics (dilation, color, friability) were worse in mice fed higher DSS concentrations. Histology revealed small intestinal disarray among all mice fed DSS, while higher DSS concentrations resulted

**Data availability statement:** All relevant data are within the manuscript and its Supporting Information files.

**Funding:** This work was supported by the following grants: Targeted Research Opportunity Grant from the Office of the Vice President of Research at Stony Brook University (ES, HH), NIH Grants R01 DK124342 (ABB), RO1 DK052230 (VWY), R01 NS088479 (LPW). The funders had no role in study design, data collection and analysis, decision to publish or preparation of the manuscript.

**Competing interests:** NO authors have competing interests.

**Abbreviations:** NEC, Necrotizing enterocolitis; DSS, dextran sodium sulfate; GI, gastrointestinal; P, postnatal; ANOVA, analysis of variance; CSS, clinical sickness score; CNS, central nervous system; BSA, Bovine serum albumin; TTBS, Tween Tris buffered saline; Ki67, antigen Kiel-67; CC3, anti-cleaved caspase-3; HRP, horseradish peroxidase; NeuN, neuronal nuclei; Iba-1, ionizing calcium binding antigen; DAPI, 4',6-diamidino-2-phenylindole; IL, interleukin; C-X-C motif, chemokine; CXCL1, ligand 1; CCL2, chemokine (C-C motif) ligand 2; IFN-γ, interferon-γ; TNF-α, tumor necrosis factor α; G-CSF, granulocyte colony stimulating factor; GM-CSF, granulocyte-macrophage colony stimulating factor; PFA/PB, paraformaldehyde/phosphate buffer; PBS, phosphate buffered solution.

in reduced small intestinal cellular proliferation and increased hepatic and systemic inflammation. In the brain, IL-2, G-CSF, and CXCL1 concentrations increased with higher DSS concentrations, and microglial branching in the hippocampus CA1 was significantly reduced in DSS-fed mice. In conclusion, we characterized a novel graded model of NEC that recapitulates the full range of NEC severities. We showed that mild NEC is sufficient to initiate neuroinflammation and microglia activation. This model will facilitate long-term studies on the neurodevelopmental effects of NEC.

## Introduction

Necrotizing enterocolitis (NEC) is the most frequent intestinal emergency among preterm newborns and accounts for ~125,000 cases per year in North America [1]. ~70% of NEC patients can be treated medically (bowel rest, antibiotics, and parenteral nutrition), but severely affected patients require surgery for necrotic or perforated bowel [2]. Using Bell's staging criteria, NEC severity is graded into three stages (mild, moderate, and severe) based on clinical parameters like abdominal distension, hemodynamic stability, radiological findings, and the necessity of surgical intervention [2]. Overall mortality for NEC ranges from 15 to 30%, whereas half of surgical NEC patients die [3–5]. In addition to gastrointestinal (GI) dysfunction, the major long-term complications of NEC are diminished growth and neurodevelopmental impairments, which are found in nearly 60% of NEC survivors [6].

The neurological outcomes following NEC range from mild developmental delays to severe cognitive dysfunction and neuromotor deficits. NEC patients have a 50% increased risk for neurodevelopmental impairments compared to gestational age-matched cohorts of premature patients [6]. These neurological symptoms are associated with changes in cross-sectional imaging including white matter lesions, cortical thinning, and loss of gray matter volume [7]. Even patients who have had stage I/II NEC experience behavioral problems and learning disabilities in school at a higher rate than premature patients without NEC [8–10].

The pathways underlying the complex relationship between GI pathology and central nervous system (CNS) changes remain incompletely understood. Previous literature has demonstrated changes in brain volume, neuronal cell numbers, inflammatory markers, and microglia activation as a direct consequence of severe NEC in animal models and human postmortem samples [7,11–14]. However, current models are limited in their ability to study the initial steps of NEC pathogenesis and cannot prospectively induce mild and severe NEC in animal models. A postmortem examination is typically required to correlate NEC disease severity with CNS changes. An animal model that can prospectively control NEC severity would be a valuable tool to study the effects of NEC severities on subsequent neurodevelopmental outcomes.

Building upon a NEC model published by Ginzel et al. in 2017 [15], we developed and characterized a graded model of NEC that allows us to control the severity of induced enterocolitis. By adding increasing concentrations of dextran sodium sulfate (DSS), an osmotic agent, to enteral gavage feedings of neonatal mice starting on

the third day of life, we demonstrated a graded response in mortality and clinical characteristics, as well as in intestinal morphology and disease appearance. We chose an experimental period of postnatal days 3–6 in mouse neurodevelopment, which corresponds with the third trimester of human brain development, thus allowing us to model how NEC affects brain development in preterm infants [16]. This novel graded neonatal NEC mouse model will advance our understanding of the initial stages of NEC, their transition to more advanced NEC stages, and the interplay between the GI, immune, and nervous systems.

## Materials and methods

### Animals

All animal experiments were performed according to a protocol approved by the Institutional Animal Care and Use Committee (IACUC) of Stony Brook University, Stony Brook, NY, in concordance with the guidelines established by the National Institutes of Health (NIH). Mice (C57BL/6, Charles River, Wilmington, MA) were bred and housed within Stony Brook University's animal care and research facilities, which are part of the Division of Laboratory Animal Resources (DLAR).

**Ethics Statement:** This research did not involve human subjects or tissue. All animal research was performed under a protocol approved by the Stony Brook University IACUC. Euthanasia of mice younger than P10 was done via decapitation with scissors; no animals older than P10 were used for this study.

### Necrotizing enterocolitis induction with dextran sodium sulfate

Enterocolitis was induced in mice, as described by Ginzel et al. (2017) [15]. All mice fed formula or DSS supplemented formula were separated from their dams and specially housed in an Ohmeda Medical Ohio® Care Plus Incubator (36.8°C) for the entire feeding duration, starting on postnatal day 3 (P3). Animals receiving the same experimental treatment were housed together, to minimize distress, in plastic bins with clean bedding within the incubator. Individual animals were marked for identification on their backs using water soluble marking pens. Starting two hours after separation from the dam, pups were fed by orogastric gavage on a heated pad every three hours for 72 hours total. Research staff were specifically trained to handle young mice (<P10) and perform orogastric gavage feeding safely. Pups were returned to the incubator immediately after feeding. Pups were initially fed with 50 μL of Esbilac formula (PetAg, Hampshire, IL) with or without DSS (Sigma-Aldrich, St Louis, MO) supplementation. Gavage volumes were increased by 10 μL daily to account for weight gain (50 μL, 60 μL, and 70 μL for days 1, 2, and 3 of feeding, respectively). Individual mice from the same litter were randomly assigned, without consideration of sex, to control or experimental groups. The experimental groups included 72-hour feeding with DSS dissolved in formula at one of the following w/v concentrations: 0.25%, 1%, 2%, or 3% DSS.

Neonatal mice fed formula alone, i.e., 0% DSS, served as controls in the characterization of this model to avoid maternal separation as a confounding factor. Furthermore, animals raised in a sterilized incubator will be exposed to a different microbial environment than that encountered in a cage housing the animals. Unlike nursing mice left with dams, formula-fed mice experience isolation from their dams, which has been shown to increase stress and anxiety and negatively impact neurodevelopment [17]. To confirm differences in development between these two groups, control formula-fed pups were compared to pups nursed by their dams. Nursing mice demonstrated an increased weight gain and differences in cytokine and chemokine profile (S1 Fig) than formula-fed mice.

Animals were monitored every three hours for signs of NEC (rectal bleeding, lethargy, and abdominal distension) and general health and behavior. Body weight was recorded every 12 hours, beginning at the initiation of feeds (0 hours). A clinical sickness score (CSS) was assessed (general appearance, response to touch, natural activity, body color) [18], every 12 hours. As per the humane endpoints defined in the IACUC protocol, animals were immediately euthanized if they exhibited apnea, rectal bleeding, lethargy, weight loss > 15% of total body weight per day, or no weight gain for two days.

Tissue samples were prepared for analysis if animals were euthanized by these indications, thus shortening the duration of the experiment for these mice. The remaining animals were euthanized after 72 hours of feeding, and tissues were harvested for histological and immunological studies. Animals in the 0%, 0.25%, and 1% DSS groups that died prior to meeting the criteria for euthanasia were eliminated from the study analysis (aside from the Kaplan-Meier survival curve, Fig 1A).

## Gastrointestinal methods

**Intestine evaluation and preparation.** Intestines were placed in cold phosphate-buffered saline (PBS) following dissection. External bowel scores were determined, as previously described by Zani et al. (2008) [18]. Bowels were scored upon color, dilation, and friability/consistency for a maximum sickness score of 9 (3 points per category). Intestines were

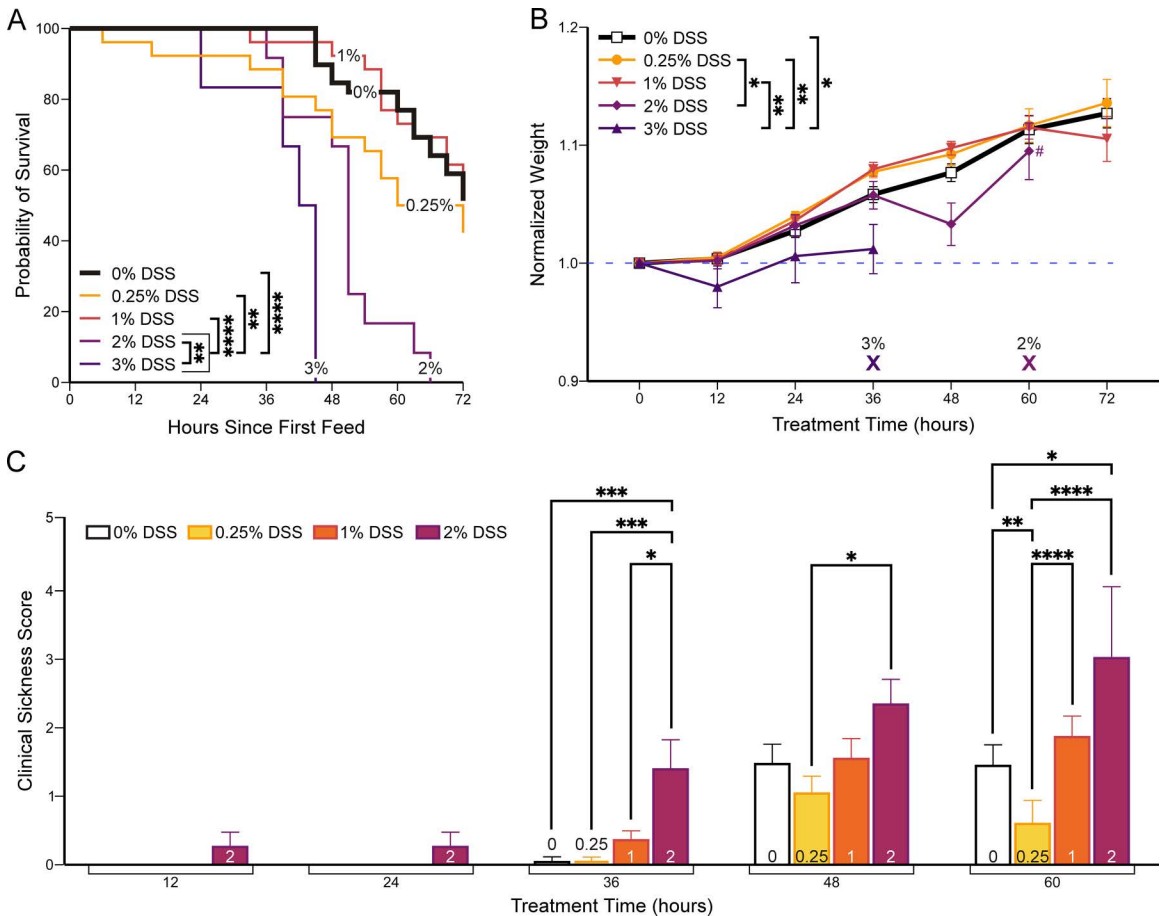

**Fig 1. Survival, weight gain, and clinical sickness scores (CSS) correlate with dextran sodium sulfate (DSS) concentration.** Starting on postnatal day 3, mice were fed either 0% (control) or increasing concentrations of DSS (0.25%, 1%, 2%, or 3%) for 72 hours (see Methods). **(A)** Kaplan-Meier survival curve. Mice fed with high concentrations of DSS (2% & 3%) had a shorter lifespan compared to lower concentrations. Pairwise survival analysis between high DSS and low DSS groups, $p < 0.0001$ (S1 Table). Number of mice: 0%, 40; 0.25%, 26; 1%, 26; 2%, 12; 3%, 6. **(B)** Normalized weight trends during protocol. X indicates the time point at which all mice died or were euthanized within that experimental group. **#** at 60 hours where only 2 mice have survived in 2% DSS (n: 12 → 2). Two-way ANOVA with Tukey's post-hoc, $p < 0.0001$ (S2 and S3 Tables). The number of mice are the same as in 1A. **(C)** CSS increased (worsened) more rapidly in animals exposed to higher DSS concentrations (see S2 Fig for scores per category). Two-way ANOVA with Tukey's post-hoc, $p < 0.0001$ (S4 and S5 Tables). Number of mice: 0%, 29; 0.25%, 26; 1% 23; 2%, 7. 3% DSS data not collected because mice were too sick to be assessed. Data presented as mean±SEM. *$p < 0.05$, **$p < 0.01$, ***$p < 0.001$, ****$p < 0.0001$.

then prepared for fixation, as described in Bialkowska et al. (2016) [19]. Mesentery and connective tissue were removed. The bowel was flushed with modified Bouin's fixative (50% ethanol/ 5% acetic acid in $dH_2O$) and cut at the ileocecal junction. Using the Swiss-rolling technique [19], the bowel was fixed overnight in 10% formaldehyde at room temperature and transferred to cold 0.1M PBS the following day. The bowel was paraffin-embedded and sliced into 5 µm sections for hematoxylin and eosin (H&E) or immunohistochemical staining. The Research Histology Core Laboratory, Department of Pathology at Stony Brook University, performed tissue processing, paraffin embedding, slicing, and H&E staining.

**Intestinal immunohistochemistry (IHC).** 5 µm intestinal paraffin slides were stained as described by Talmasov et al. (2015) [20]. Sections were deparaffinized in xylene, incubated in 2% hydrogen peroxide in methanol for 30 minutes, rehydrated in an ethanol gradient, and then treated with 10 mM Na citrate buffer, pH 6.0, at 120°C for 10 min in a pressure cooker. The slides were then blocked with 5% bovine serum albumin (BSA) in wash solution (0.01% Tween 20, 1X Tris-buffered PBS (TTBS)) and incubated with primary antibodies (Table 1; anti-Ki67, anti-cleaved caspase-3) overnight at 4°C. Slides were washed with TTBS, incubated with a secondary antibody horseradish-peroxidase (HRP) probe from the MACH 3 Rabbit HRP Polymer Detection kit (Biocare Medical, Pacheco, CA), washed with TTBS again, and treated with a tertiary antibody red HRP polymer from the same kit. After the final TTBS washes, color was developed using the Betazoid DAB Chromogen kit (Biocare Medical, Pacheco, CA) and counterstained with hematoxylin solution, Gill (Sigma-Aldrich, St Louis, MO). Slides were dehydrated using alcohol and xylene gradient before mounting.

**Slide scanner light microscopy.** Stained intestinal images were acquired on an Olympus VS120 Virtual Slide Microscope (Olympus Corporation, Japan) at 20x magnification. Images were processed using Olympus Viewer software, and regions of maximal necrosis or proliferation (based on Ki-67 staining) were saved at 200x magnification (10x zoom of the original scan taken at 20x magnification). These 200x magnification small intestinal images were selected for cell count analysis.

**Analysis of intestinal tissue.** H&E-stained intestinal sections were analyzed as per Tanner et al. (2015) and Liu et al. (2022) [21,22]. A minimum of six animals were used for each condition for enterocolitis grading. Intestinal pathology was scored based on observation of the region with the worst lesion. Briefly, the scores were as follows: 0 for intact villi and crypts, 1 for superficial epithelial sloughing, 2 for mild villous or crypt necrosis, 3 for complete villous or crypt necrosis, and 4 for transmural necrosis [21]. For Ki-67 counts, 4–10 crypts were identified per 200x magnification image (3 high-power fields were analyzed per animal). Ki-67-positive or caspase-positive cells, if present, were counted for each crypt. A minimum of three animals were analyzed for each condition for Ki-67 staining.

## Cytokine and chemokine preparation and analysis

**Tissue preparation for cytokine and chemokine measurements.** Organ tissue samples (liver and cortex) were collected into sterile microcentrifuge tubes and weighed before being homogenized in sterile endotoxin-free saline using

**Table 1. Primary (1°) antibodies used for IHC of intestine (paraffin) and brain (frozen) slices.**

| 1° antibody | Target | Species | Dilution | Company | Catalog # |
|---|---|---|---|---|---|
| *Intestine (paraffin)* | | | | | |
| Kiel 67 (Ki-67) | Proliferating cells | Rabbit | 1:200 | Biocare Medical | CRM325B |
| Cleaved caspase-3 (CC3) | Apoptotic cells | Rabbit | 1:400 | Cell Signaling | Ab#9661 |
| *Brain (frozen)* | | | | | |
| Neuronal nuclear protein (NeuN) | Mature neurons | Mouse | 1:100 | MilliporeSigma | MAB377A5 |
| Ionized calcium binding adaptor molecule-1 (IBA1) | Microglia, macrophages | Rabbit | 1:500 | Wako | 019-19741 |

Primary antibodies used for immunohistochemical staining of either paraffinized intestinal slices or fixed frozen brain slices (see *Methods*). For the intestines, Ki-67 and CC3 were used independently in different slices, whereas, for the brain, NeuN and IBA1 were co-stained, with DAPI, in the same slices.

sterile 2.3 mm zirconia/silica beads (Biospec Products Inc.; Bartlesville, OK), as previously described by Speer et al. (2020) [23]. Samples were then centrifuged at 13,000 g for 10 min at 4°C. Supernatants were harvested and stored at −80°C for subsequent cytokine and chemokine measurements.

**Cytokine and chemokine analysis.** Concentrations of 16 cytokines and chemokines (IL-1α, IL-1β, IL-2, IL-6, IL-10, IL-12(p40), IL-12(p70), IL-17, CXCL1, CCL2, CCL3, CCL4, G-CSF, GM-CSF, IFN-γ, TNF-α) in tissue homogenate supernatants and plasma were measured with Bio-Plex Pro magnetic multiplex assays (Bio-Rad; Hercules, CA) and analyzed on the Bio-Plex 200 system with Bio-Plex Manager 5.0 software (Bio-Rad). Plasma was isolated from whole blood that was spun at 1,000G for 10 minutes. Results were expressed as pg per mL (plasma) or pg/mg protein (liver, cortex) cytokine or chemokine concentration for supernatants of organ tissue homogenates. Duplicate technical replicates were used for all immunological studies. Protein concentrations in supernatant tissue homogenate samples were determined with the Bradford method (Bio-Rad Laboratories; Richmond, CA) and measured on a Spectramax 190 Plate Reader (Molecular Devices LLC; San Jose, CA), as previously described [23].

## Brain analysis methods

**Brain immunofluorescence.** Whole brains were dissected and fixed in 4% paraformaldehyde/phosphate buffer (PFA/PB) at 4°C overnight and then moved to PBS with 30% sucrose/phosphate-buffered saline (PBS). After removing the cerebellum, brains were embedded in an optimal cutting temperature (OCT) compound (Fisher Scientific, Hampton, NH). 20 µm coronal sections were made using a cryostat (Leica, Germany). The sections were washed in 0.3% Triton-X100/PBS three times, blocked with 10% goat serum in 0.3% Triton-X100/PBS, and incubated with primary antibodies (Table 1; anti-IBA-1 (rabbit), anti-NeuN conjugated to Alexa-Fluor 555) overnight at 4°C. Next, sections were washed three times in 0.3% Triton-X100/PBS and treated with the secondary antibody goat anti-rabbit Alexa Fluor 488 (ThermoFisher, Waltham, MA; 1:1000) overnight at room temperature. After three washes in 0.3% Triton-X100/PBS, the slides were mounted in DAPI Fluoromount (SouthernBiotech, Birmingham, AL), left to dry overnight, and sealed.

**Confocal microscopy and image acquisition.** Images were captured with an Olympus FV1000 confocal microscope (Olympus Corporation, Japan) with 30x (silicon immersion) and 60x (oil immersion) lenses to image the CA1 hippocampal region of stained sections. 30x magnification images were taken first for neural cell counts. Subsequently, to obtain higher resolution visualization of microglia for Sholl analysis, regions on the 30x magnification images were chosen using a random number generator, to avoid selection bias, for reimaging at 60x magnification. Z-stack images (18-22 stacks, each 1 µm thick) were captured using Olympus FluoView software. Images were subsequently processed and analyzed with Fiji is Just ImageJ (Fiji) software [24].

**Image analysis of brain tissue.** All experimenters were blinded to the treatment conditions during image analysis. At least two independent individuals analyzed identical samples. The results were compared to ensure consistency of analysis, and final counts were averaged between two or more individuals. If counts between individuals varied more than 15%, images were re-analyzed. IBA-1 and DAPI co-stained cells were identified as microglia, whereas neurons were identified by their NeuN and DAPI co-staining. DAPI, IBA-1, and NeuN cells were counted for each high-powered field. Cell proportions per image were calculated by dividing the number of microglia or neurons by the number of DAPI-positive cells. Additionally**,** microglia morphology was analyzed for branching complexity via Sholl analysis [25]. All microglia dendrites (imaged at 60x magnification) were manually traced and skeletonized. The branching complexity of the skeletonized cells was assessed using the Sholl Analysis plugin for Fiji software with the following settings: 5 µm starting radius, 70 µm ending radius, and 5 µm step size [26]. These settings instructed the software to count the number of intersecting branches at each circle drawn around the cell soma. The first circle was set at a distance of 5 µm from the cell soma, then each concentric circle was set every 5 µm away from the previous one until the last circle was drawn at 70 µm from the cell soma.

## Statistical analysis

Statistical analysis and graphing of results were performed using Excel and GraphPad Prism software v. 9 and 10 (GraphPad Software; San Diego, CA). P-values less than 0.05 were considered statistically significant. A survival curve log-rank test was performed for mortality. Two-way analysis of variance (ANOVA) tests were used for weight data, behavior scores, intestinal pathology, and Sholl analysis of microglia. One-way ANOVAs were employed for bowel scores and all cell counts (both for the intestines and the brain). All ANOVA tests were verified by a post-hoc Tukey analysis for all data pair comparisons. Simple linear regression analysis was also conducted for external bowel scores and intestinal cellular proliferation counts. For cytokine and chemokine results, log transformations, using the equation $y = \log(y)$, were first applied to meet the normality assumptions as indicated. A simple linear regression model was then used to fit the outcome values.

## Results

Enteral administration of formula supplemented with 3% DSS, an osmotic agent, to neonatal mice elicited severe entero-colitis characteristic of NEC [15]. Taking advantage of this general approach, we fed neonatal mice either formula alone (0% DSS control) or formula supplemented with DSS at a specific concentration (0.25%, 1%, 2%, or 3% DSS) for 72 hours to determine if we could titrate the severity of NEC induced to mimic NEC stages seen in neonatal human patients. We first assessed clinical characteristics, such as mortality, weight changes, and sickness scores.

### Increasing DSS supplementation results in higher mortality rate, decreased weight gain, and earlier abnormal clinical sickness scores

Approximately 50% of all experimental mice survived the feeding protocol. Mice either died between feeds or met humane endpoints for euthanasia, as per the IACUC protocol (*see Methods*). Only mice who were euthanized were analyzed. The total number of mice used for each experimental treatment is as follows: 0% DSS, 40 mice; 0.25% DSS, 26 mice; 1% DSS, 26 mice; 2% DSS, 12 mice; 3% DSS, 6 mice. All mice in the 2% DSS and 3% DSS groups, as well as a quarter of the mice in the 1% DSS group, met the determined humane endpoints and were euthanized. Another quarter of 1% DSS mice, as well as about half of 0% DSS and 0.25% DSS mice, were found dead or died prematurely, the cause of which could have been bowel perforation, sepsis/non-NEC infections, or handling and stress. Mice in the 0%, 0.25%, and 1% DSS groups had comparable Kaplan-Meier survival curves (Fig 1A), which all differed significantly from survival curves in the 2% and 3% DSS groups ($p < 0.01$, pairwise Kaplan-Meier survival analysis between the three low DSS groups and the two high DSS groups, survival curve log-rank). All mice in the 2% DSS group were euthanized before 72 hours, and mice in the 3% DSS group were euthanized before 48 hours ($p < 0.01$, survival curve log-rank, for 2% versus 3% DSS).

To determine if DSS concentration would affect growth during the feeding period, we weighed mice every 12 hours. Higher concentrations of DSS supplementation (2% and 3%) resulted in less weight gain when compared to mice fed with lower DSS concentrations (0%, 0.25%, 1%). Mice fed lower DSS concentrations steadily increased their weight (Fig 1B). The final mean normalized weights at 72 hours are summarized in S2 Table. There was no statistically significant difference in overall weight gain among low DSS concentration groups (0%, 0.25%, 1%) ($p = 0.354$, two-way ANOVA with Tukey's post-hoc). Between the 48- and 60-hour time points, only 20% of the 2% DSS mice survived, which explains the sudden deviation in the mean normalized weight. The 3% DSS mice deteriorated rapidly; therefore, we omitted 3% DSS-fed mice from subsequent studies and used 2% DSS to represent severe enterocolitis.

Throughout the feeding protocol, we used a CSS [18] to assess neonatal mouse behavior and health. Mice in the 2% DSS group began exhibiting signs of illness – concerning appearance, activity, and response to touch – as early as 12 hours of DSS administration; the CSS in these mice continued to increase over time (Fig 1C). In contrast, mice exposed to lower concentrations of DSS (0%, 0.25%, 1%) began exhibiting abnormal behavior only after 36 hours of feeding. 2% DSS-fed mice demonstrated significantly higher clinical sickness scores compared to all other DSS conditions after 36

hours ($p < 0.05$, two-way ANOVA with Tukey's post-hoc, S5 Table). Overall, increased DSS supplementation resulted in significantly decreased survival, decreased weight gain, and worse clinical sickness scores in mice.

## DSS feeds induced enterocolitis in neonatal mice

NEC causes intestinal inflammation, inflammatory cell recruitment, and villous sloughing. This process progresses through the muscularis layer and can eventually result in transmural necrosis and perforation [21]. After three days of feeding, we evaluated gross pathology and histology (Fig 2A) of the intestines to determine if increasing concentrations of DSS resulted in worsening NEC and GI damage. The external bowel score (Fig 2B, S3 Fig) assesses gut color, dilation, and consistency or friability [18]. We found that the external bowel score was significantly correlated with increasing DSS concentrations (slope: 2.41 with $p < 0.0001$).

Complementing the gross pathology, H&E staining of DSS-treated intestines (Fig 2A: 2nd and 3rd rows) also indicated the presence of enterocolitis with derangement of villous organization, intestinal dilatation with patchy necrosis and perforation. Consistent with previous literature [15], enteral DSS administration primarily caused enterocolitis in this newborn mouse model (Fig 2C, S8 Table), which contrasts with the colitis observed in older mice treated with DSS [27]. We scored the intestinal pathology based on the worst pathology observed in the slide. All mice exposed to DSS demonstrated enterocolitis throughout, in both the proximal and distal/terminal segments of the small intestines. Intestines from mice fed 0.25%, 1%, and 2% DSS scored significantly worse than those from 0% DSS-fed mice ($p < 0.0001$, $p = 0.026$ and $p = 0.0008$, respectively; S9 Table). Animals administered 0.25% and 2% DSS displayed minor colitis, which was not significantly different from controls. Following these findings, we performed immunohistochemistry to determine if enterocolitis was associated with cellular apoptosis or proliferation abnormalities.

## Increased DSS exposure suppresses intestinal cell proliferation without promoting apoptosis

We assessed the intestines of DSS-exposed and formula-fed (control) mice for the presence of cellular apoptosis or proliferation by staining for CC3 or Ki-67 (Fig 2A: bottom row), respectively. For all experimental groups, small intestinal crypts did not show any positive staining for CC3 (S4 Fig). Though previous studies have demonstrated an increase in apoptosis, we believe not seeing a similar change in our animals is unsurprising, as we had to sacrifice animals as soon as they developed bloody diarrhea, according to our IACUC protocol. This time point may have been too soon for apoptosis to occur. In contrast, Ki-67 positive (Ki-67+) staining of small intestinal crypts in 2% DSS-fed mice was significantly decreased compared to the staining in 0% or 0.25% DSS-fed pups (Fig 2D, S10 and S11 Tables). Simple linear regression analysis of the Ki-67 + cell counts demonstrated that cell proliferation was negatively correlated with DSS concentration (slope: −2.94, $p = 0.0009$).

## Higher DSS concentrations in feeds led to hepatic and systemic inflammation

Overall, we observed that higher concentrations of DSS resulted in worsening clinical signs, intestinal pathology, and diminished intestinal cell proliferation. NEC causes local intestinal inflammation accompanied by cytokine and chemokine production that propagates to a systemic response [28]. We therefore measured cytokine and chemokine levels in the liver to determine if these inflammatory mediators correlate with the DSS concentration.

Since the liver is the first major filtration organ downstream of the intestines, we used liver data as a measure of gastrointestinal cytokine and chemokine production. Notably, all cytokine and chemokine concentrations in liver tissue positively correlated with increasing supplementation of DSS (Figs 3 and 4, Table 2). Chemokine production induces the release and recruitment of immune cells to the peripheral tissues. We performed the same analysis on blood plasma; several cytokine and chemokine levels were below the threshold of detection. In plasma, concentrations of IL-12, G-CSF, CXCL1, CCL4, and IL-10 correlated with increasing DSS supplementation (Fig 5, Table 2).

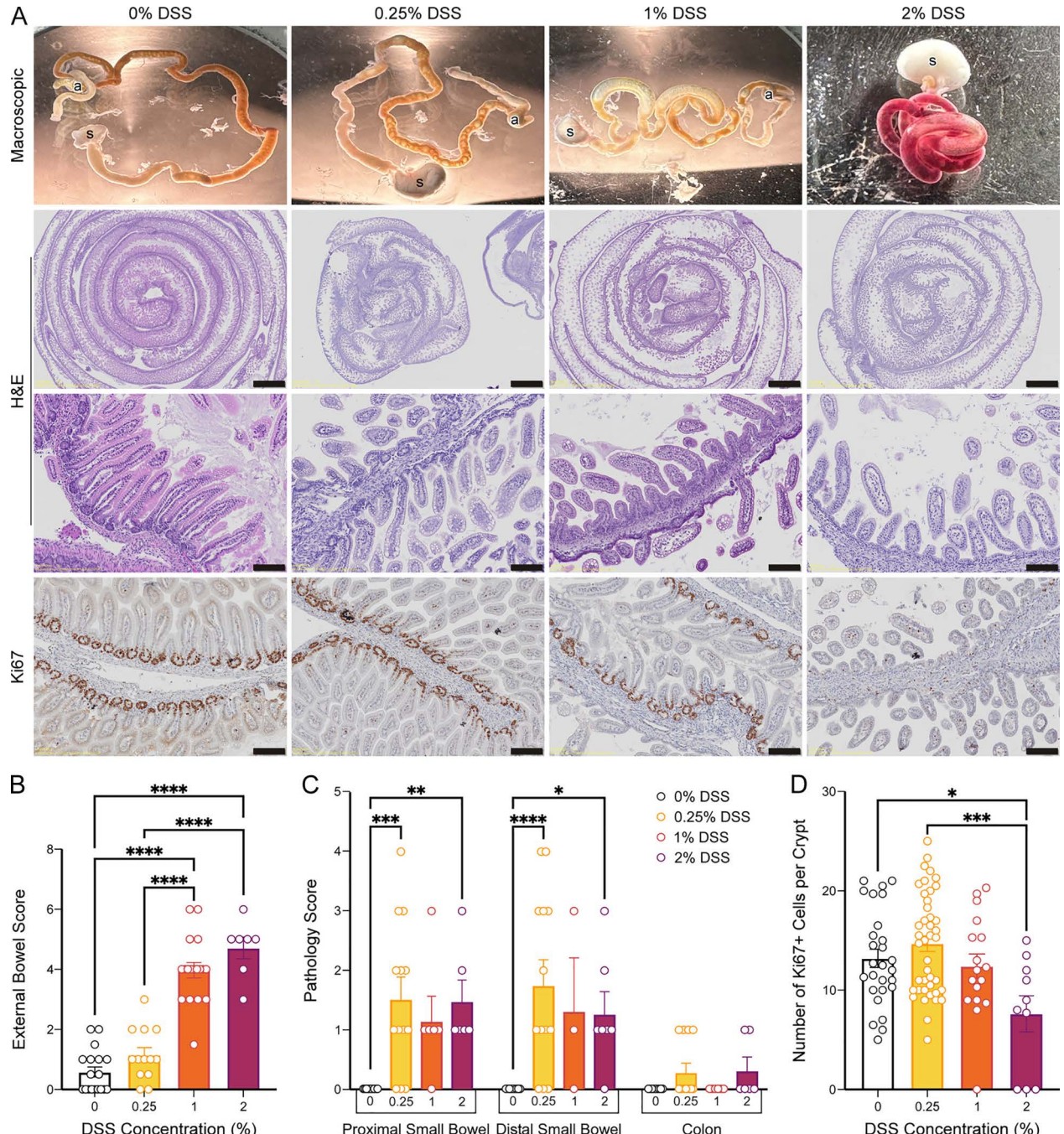

**Fig 2. Enterocolitis severity increases with DSS concentration.** Increased DSS concentration led to increased external bowel score, disruption of intestinal morphology, and decreased Ki-67+cells. **(A)** Representative images of the entire intestinal tract macroscopically (1st row), hematoxylin- and eosin-stained (H&E) intestinal samples imaged at 10x (2nd row, scale 1mm) or at 200x magnification (3rd row, scale 100 μm) or of Ki-67 immunohisto-chemistry at 200x magnification (4th row, scale 100 μm). **(B)** External bowel severity scores are higher (worse) at increasing DSS concentrations, 1 & 2% (see S3 Fig for scores per category). One-way ANOVA with Tukey's post-hoc, p < 0.0001 (S6 and S7 Tables). Number of mice: 0%, 18; 0.25%, 13; 1%, 18; 2%, 7. **(C)** Enterocolitis pathology, which was only present in DSS groups, was primarily localized to the small intestine. Two-way ANOVA with Tukey's post-hoc, p < 0.0001 (S8 and S9 Tables). Number of mice: 0%, 7; 0.25%, 7; 1%, 6; 2%, 3. **(D)** Cellular proliferation, as measured by number of Ki-67-positive cells within small intestinal crypts, decreased significantly in 2% DSS. One-way ANOVA with Tukey's post-hoc, p < 0.0001 (S10 and S11 Tables). Number of crypts: 0%, 27; 0.25%, 42; 1%, 16; 2%, 7. Data presented as mean±SEM. *p < 0.05, **p < 0.01, ***p < 0.001, ****p < 0.0001.

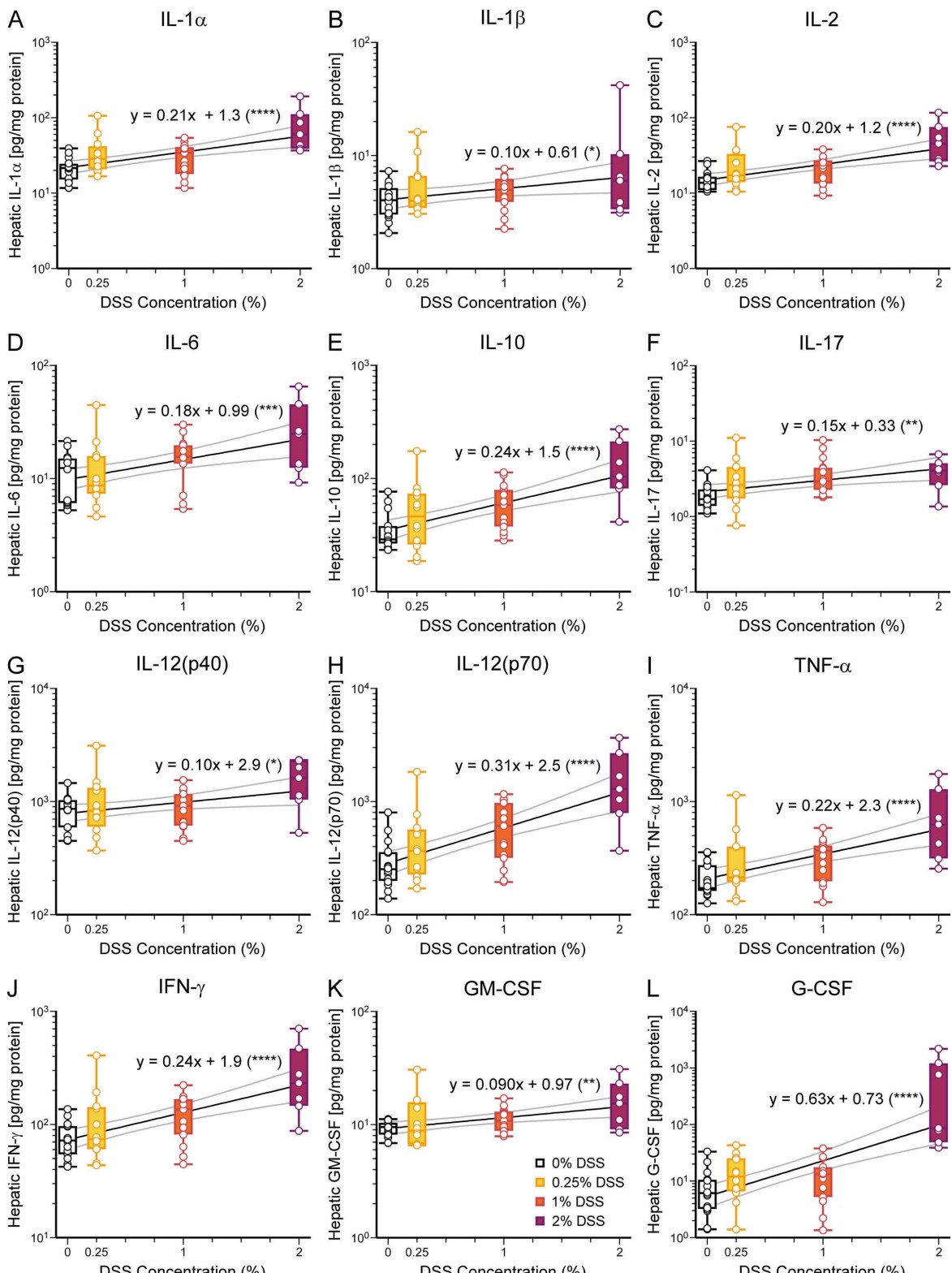

**Fig 3. Concentrations of all tested hepatic cytokines positively correlate with DSS concentration. (A-H)** In the liver, cytokine concentrations are all positively correlated with DSS concentrations. Cytokines tested include: **(A)** IL-1α, $p < 0.0001$, **(B)** IL-1β, $p = 0.033$, **(C)** IL-2, $p < 0.0001$, **(D)** IL-6,

*p* = 0.0009, **(E)** IL-10, *p* < 0.0001, **(F)** IL-17, *p* = 0.0035, **(G)** IL-12(p40), *p* = 0.020, **(H)** IL-12(p70), *p* < 0.0001, **(I)** TNF- α, *p* < 0.0001, **(J)** IFN-γ, *p* < 0.0001, **(K)** GM-CSF, *p* = 0.0024, and **(L)** G-CSF, *p* < 0.0001. Data are presented as boxplots showing min-max. Slope and y-intercept with confidence intervals are plotted; equations are included for each regression analysis. *p*-values are calculated for t statistic of slope (coefficient of DSS concentration) using simple linear regression with log-transformation of y values (see Table 2). *p* < 0.05, **p* < 0.01, ***p* < 0.001, ****p* < 0.0001. Number of mice: 0%, 16; 0.25%, 12; 1%, 15; 2%, 7.

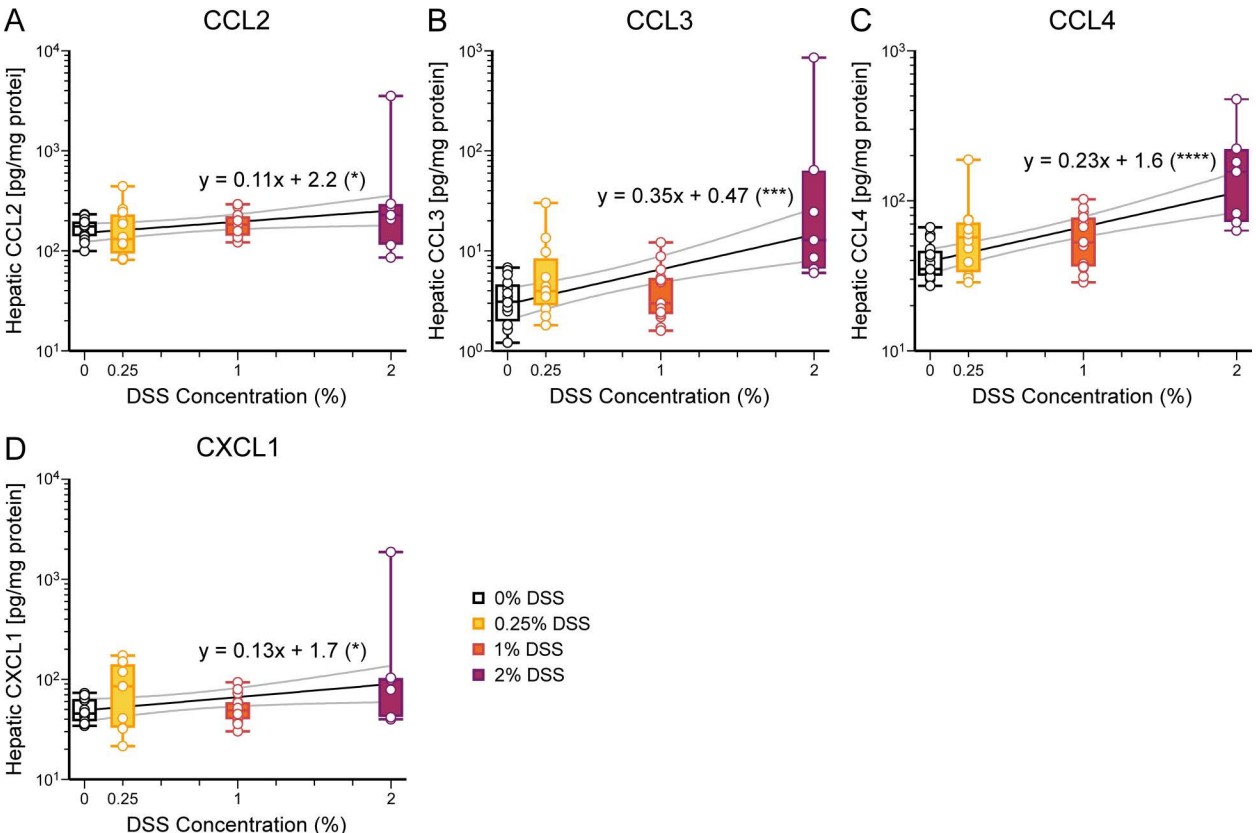

**Fig 4. Concentrations of hepatic chemokines positively correlate with DSS concentration. (A-H)** In the liver, chemokine concentrations are positively correlated with DSS concentrations. Chemokines evaluated include: **(A)** CCL2, *p* = 0.027, **(B)** CCL3, *p* = 0.0002, **(C)** CCL4, *p* < 0.0001, and **(D)** CXCL1, *p* = 0.031 (see Table 2). Results displayed and analyzed as in Fig 3. *p* < 0.05, ***p* < 0.001, ****p* < 0.0001. Number of mice: 0%, 16; 0.25%, 12; 1%, 15; 2%, 7.

In summary, increasing concentrations of DSS resulted in worsening clinical phenotypes, worsening enterocolitis, and increasing inflammation in liver tissue and plasma, as assayed by cytokines and chemokines. Given the strong association between the DSS concentration and NEC severity, we next used this model to study the neurological sequelae of NEC in mice.

## NEC induced neuroinflammation

Neurodevelopmental impairment is a significant long-term morbidity associated with NEC [6]. Even patients who have experienced milder forms of NEC are at a higher risk of cognitive disability compared to age-matched premature infants without NEC [29]. We therefore measured cytokine and chemokine concentrations in the brains of DSS-treated mice.

**Table 2. Linear regression with log-transformation analysis of cytokine and chemokine concentrations in liver tissue, blood plasma, and brain tissue.**

| | Cytokine | Slope (m) | Y-intercept (b) | R-squared | F-statistic | P-value |
|---|---|---|---|---|---|---|
| **Liver** | IL-1α | 0.21 | 1.3 | 0.30 | 21 | *<0.0001* |
| | IL-1β | 0.098 | 0.61 | 0.091 | 4.8 | *0.033* |
| | IL-2 | 0.20 | 1.2 | 0.33 | 24 | *<0.0001* |
| | IL-6 | 0.18 | 0.99 | 0.21 | 13 | *0.0009* |
| | IL-10 | 0.24 | 1.5 | 0.36 | 26 | *<0.0001* |
| | IL-12(p40) | 0.098 | 2.9 | 0.11 | 5.8 | *0.020* |
| | IL-12(p70) | 0.31 | 2.5 | 0.41 | 33 | *<0.0001* |
| | IL-17 | 0.15 | 0.33 | 0.16 | 9.4 | *0.0035* |
| | G-CSF | 0.63 | 0.73 | 0.42 | 34 | *<0.0001* |
| | GM-CSF | 0.95 | 0.97 | 0.18 | 10 | *0.0024* |
| | IFN-γ | 0.24 | 1.9 | 0.35 | 26 | *<0.0001* |
| | CXCL1 | 0.13 | 1.7 | 0.093 | 4.9 | *0.031* |
| | CCL2 | 0.11 | 2.2 | 0.098 | 5.2 | *0.027* |
| | CCL3 | 0.35 | 0.47 | 0.26 | 17 | *0.0002* |
| | CCL4 | 0.23 | 1.6 | 0.37 | 28 | *<0.0001* |
| | TNF-α | 0.22 | 2.3 | 0.33 | 23 | *<0.0001* |
| **Blood Plasma** | IL-1α | −0.040 | 1.4 | 0.0011 | 0.046 | *0.83* |
| | IL-1β | *below detection threshold* | | | | |
| | IL-2 | *below detection threshold* | | | | |
| | IL-6 | 0.45 | 0.22 | 0.078 | 3.5 | *0.070* |
| | IL-10 | 0.45 | 0.21 | 0.34 | 21 | *<0.0001* |
| | IL-12(p40) | 0.27 | 3.2 | 0.17 | 8.2 | *0.0066* |
| | IL-12(p70) | *below detection threshold* | | | | |
| | IL-17 | *below detection threshold* | | | | |
| | G-CSF | 0.64 | 2.2 | 0.22 | 12 | *0.0016* |
| | GM-CSF | *below detection threshold* | | | | |
| | IFN-γ | *below detection threshold* | | | | |
| | CXCL1 | 0.63 | 0.84 | 0.22 | 11 | *0.0016* |
| | CCL2 | *below detection threshold* | | | | |
| | CCL3 | 0.57 | 0.10 | 0.088 | 3.9 | *0.054* |
| | CCL4 | 0.63 | 1.2 | 0.18 | 9.1 | *0.0045* |
| | TNF-α | *below detection threshold* | | | | |
| **Brain** | IL-1α | 0.17 | 0.32 | 0.11 | 3.5 | *0.074* |
| | IL-1β | 0.084 | −0.30 | 0.046 | 1.4 | *0.25* |
| | IL-2 | 0.18 | 0.014 | 0.13 | 4.5 | *0.043* |
| | IL-6 | 0.083 | −0.10 | 0.018 | 0.54 | *0.47* |
| | IL-10 | 0.12 | 0.21 | 0.073 | 2.3 | *0.14* |
| | IL-12(p40) | 0.0018 | 2.4 | 0.000014 | 0.00042 | *0.98* |
| | IL-12(p70) | 0.044 | 1.2 | 0.0039 | 0.11 | *0.75* |
| | IL-17 | 0.12 | −0.70 | 0.093 | 3.0 | *0.096* |
| | G-CSF | 0.49 | 0.45 | 0.27 | 11 | *0.0030* |
| | GM-CSF | 0.17 | 0.28 | 0.10 | 3.1 | *0.089* |
| | IFN-γ | 0.15 | 0.62 | 0.085 | 2.7 | *0.11* |
| | CXCL1 | 0.21 | 1.2 | 0.27 | 10 | *0.0035* |
| | CCL2 | 0.11 | 1.6 | 0.039 | 1.2 | *0.29* |

*(Continued)*

**Table 2.** (Continued)

| Cytokine | Slope (m) | Y-intercept (b) | R-squared | F-statistic | P-value |
|---|---|---|---|---|---|
| CCL3 | −0.13 | 0.81 | 0.077 | 2.4 | *0.13* |
| CCL4 | 0.091 | 0.78 | 0.019 | 0.57 | *0.46* |
| TNF-α | 0.0059 | 0.97 | 0.00018 | 0.0051 | *0.94* |

Cytokine and chemokine concentrations were first log-transformed using the equation y = log(y) and then analyzed using simple linear regression. The results of the linear regression analysis are summarized in the table and graphically displayed in Figs 3 and 4 for liver proteins, Fig 5 and S5 Fig for plasma proteins, and Fig 6 and S6 and S7 Figs for brain proteins. The slopes of the linear regression lines were statistically compared to the line y = 0, using a two-tailed t-test, to determine if there is a significant correlation. Significant *p-values* (< 0.05) are in **bold**.

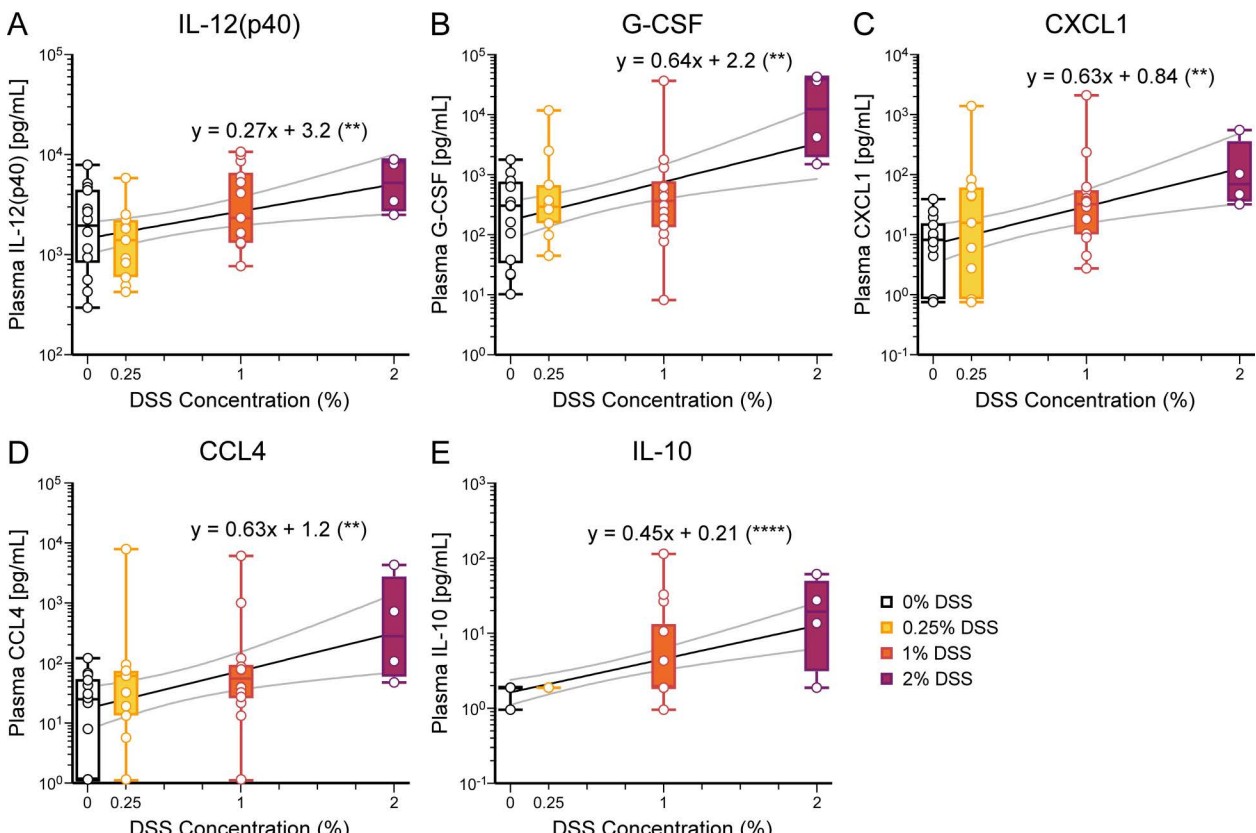

**Fig 5. Selected blood plasma cytokine and chemokine concentrations correlate with DSS concentrations. (A-E)** In blood plasma, concentrations of **(A)** IL-12(p40), *p* = 0.0066, **(B)** G-CSF, *p* = 0.0016, **(C)** CXCL1, *p* = 0.0016, **(D)** CCL4, *p* = 0.0045, and **(E)** IL-10, *p* < 0.0001, are positively correlated with DSS concentrations (see Table 2 and S5 Fig). Results displayed and analyzed as in Fig 3. **\*\****p* < 0.01, \*\*\*\**p* < 0.0001. Number of mice: 0%, 14; 0.25%, 11; 1%, 14; 2%, 4.

We measured the same array of cytokines and chemokines in brain tissue (Fig 6, S6 and S7 Figs). Overall, the concentrations of cytokines and chemokines measured in the brain were lower than those measured in the liver. Two cytokines (IL-2, G-CSF) and one chemokine (CXCL-1) increased with DSS concentration (Fig 6, Table 2). Therefore, increasing the severity of NEC induced by DSS resulted in increasing concentrations of a wide array of inflammatory cytokines and chemokines in the liver and plasma but only a small subset in the brain.

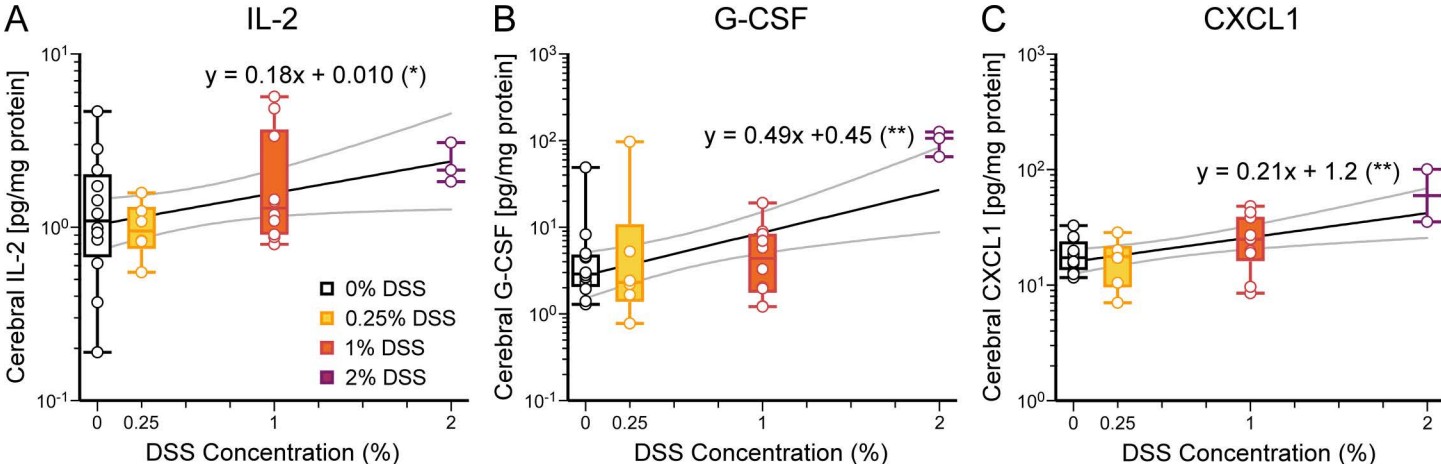

**Fig 6. Cerebral cytokines IL-2 and G-CSF and chemokine CXCL1 concentrations positively correlate with DSS concentrations. (A-C)** In the brain, concentrations of **(A)** IL-2, $p = 0.043$, **(B)** G-CSF, $p = 0.0030$, and **(C)** CXCL1, $p = 0.0035$, are positively correlated with DSS concentrations (see Table 2 and S6 and S7 Figs). Results displayed and analyzed as in Fig 3. *$p < 0.05$, **$p < 0.01$. Number of mice: 0%, 12; 0.25%, 6; 1%, 10; 2%, 3.

### NEC increased microglia activation, but not microglia number, in the hippocampus CA1 region

Next, we determined the impact of graded NEC on central nervous system anatomy, specifically for neurons and microglia, the resident immune cells of the central nervous system (S8 Fig). There was a slight increase in neuron proportion between 1% to 2% DSS supplemented mice ($p = 0.027$, One-way ANOVA with Tukey's post-hoc); otherwise, cell counts across groups were comparable. For microglia, there was a statistically significant difference only between 0.25% and 1% DSS conditions (0.25 vs. 1%, $p = 0.018$, One-way ANOVA with Tukey's post-hoc).

Microglia retract their filopodia upon activation and assume a more amoeboid shape [30,31]. To assess microglia morphology, we used Sholl analysis on individual IBA-1 positive microglia imaged at 60x magnification (Figs 7A and 7B) [25]. Sholl analysis counted the number of branching intersections at set circular distances from the cell soma, starting at 5 µm from the cell soma and extending every 5 µm until the last distance of 70 µm from the cell soma. Branching intersections peaked at 10–15 microns from the cell soma and tapered at 70 microns (Fig 7C, Table 3). Microglia from control (0% DSS) mice demonstrated significantly greater branching patterns compared to all mice with DSS supplementation (0.25, 1, 2%) ($p < 0.0001$, Two-way ANOVA with Tukey's post-hoc). Given that the branching complexity of microglia is inversely related to their activation level [31], these results indicate that DSS-fed mice displayed more activated microglia than those in control (0% DSS-fed) mice.

### Discussion

Necrotizing enterocolitis is a major cause of acute and long-term morbidity in premature infants [2]. Novel approaches are needed to understand this multifactorial disease process [32,33]. In patients, NEC causes symptoms of increasing severity. These patients develop abdominal distension and mild enterocolitis, which can then progress to fulminant sepsis, intestinal necrosis, and death. We have characterized a neonatal mouse model where NEC severity can be prospectively controlled, as demonstrated by clinical and behavioral data, intestinal pathology, and cytokine and chemokine measurements. We demonstrated that enterocolitis induced neuroinflammation characterized by microglial activation and increased expression of cerebral cytokines and chemokines. These data corroborate clinical studies where NEC patients demonstrate an increased risk of neurocognitive impairment [6].

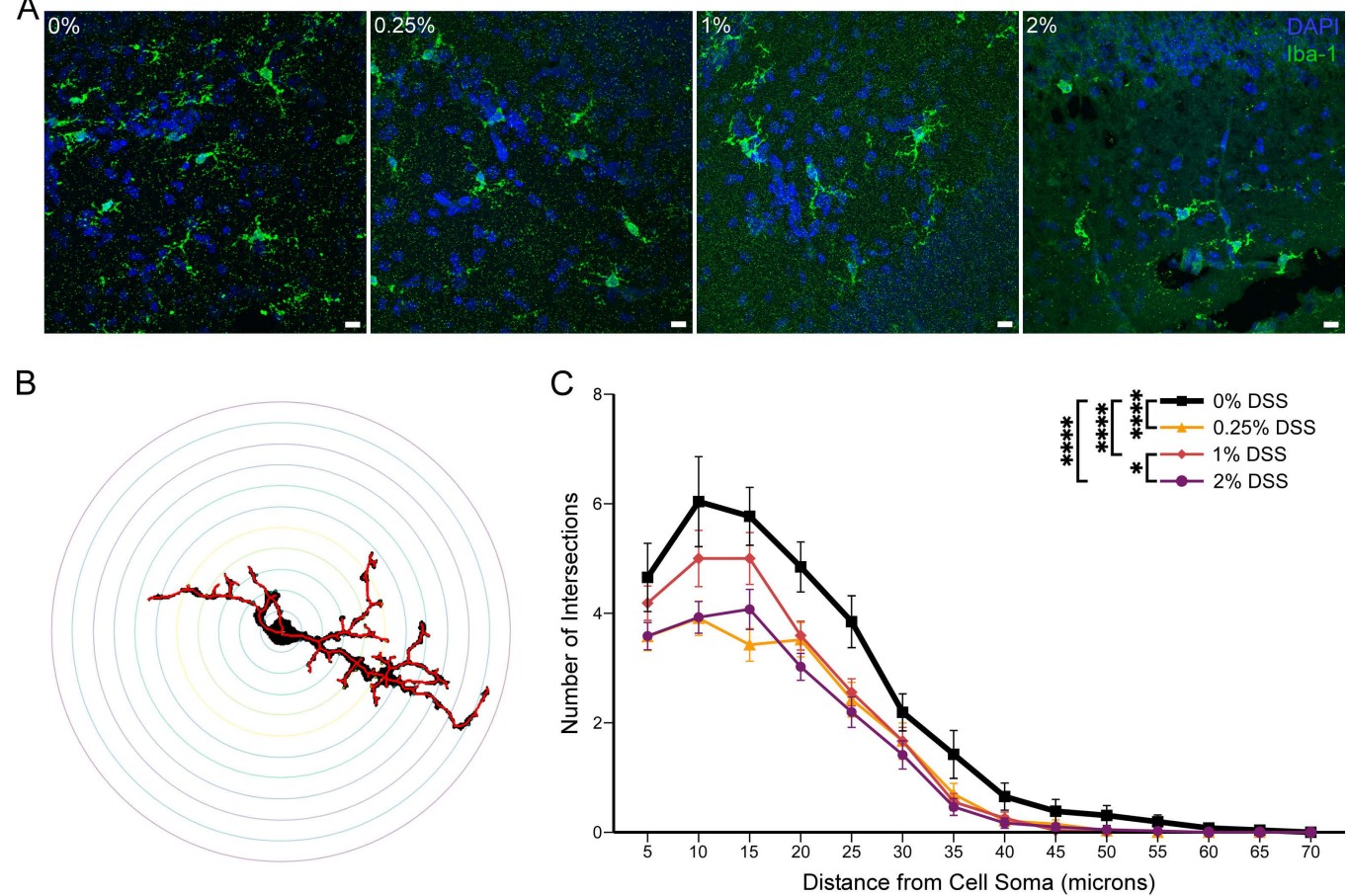

**Fig 7. DSS supplementation at any concentration activates microglia in the CA1 hippocampal region. (A)** Representative immunohistochemical images (60x magnification) of microglia in the CA1 hippocampal region. Slices are stained blue for DAPI and green for microglia. Scale bar = 10 μm. **(B)** Representative Sholl analysis tracing, with skeletonized trace in red, of a microglial cell identified by Iba-1 (0% DSS). **(C)** Sholl analysis. All DSS concentrations showed a reduced number of branching intersections (more activation) than the 0% DSS-fed mice. Two-way ANOVA with Tukey's post-hoc, $p < 0.0001$ (see Table 3). Number of microglia: 0%, 26; 0.25%, 33; 1%, 27; 2%, 41. Data presented as mean ± SEM. *$p < 0.05$, ****$p < 0.0001$.

**Table 3. Sholl analysis comparisons at different distances from microglial cell soma.**

| Comparison | Distance from Cell Soma (microns) | | | | | | | | |
|---|---|---|---|---|---|---|---|---|---|
| | 5 | 10 | 15 | 20 | 25 | 30 | 35 | 40 | Overall |
| 0% vs 0.25% DSS | *0.021* | *<0.0001* | *<0.0001* | *0.0022* | *0.0009* | 0.50 | 0.21 | 0.64 | *<0.0001* |
| 0% vs 1% DSS | 0.63 | *0.041* | 0.20 | *0.0078* | *0.0056* | 0.54 | 0.12 | 0.75 | *<0.0001* |
| 0% vs 2% DSS | *0.015* | *<0.0001* | *<0.0001* | *<0.0001* | *<0.0001* | 0.13 | *0.037* | 0.53 | *<0.0001* |
| 0.25% vs 1% DSS | 0.35 | *0.017* | *0.0001* | >0.99 | 0.98 | >0.99 | 0.98 | >0.99 | 0.089 |
| 0.25% vs 2% DSS | >0.99 | >0.99 | 0.21 | 0.46 | 0.90 | 0.87 | 0.90 | >0.99 | 0.97 |
| 1% vs 2% DSS | 0.33 | *0.013* | *0.044* | 0.38 | 0.74 | 0.89 | 0.99 | 0.99 | *0.020* |

Sholl analysis was used to quantify dendritic branching complexity of microglia, with a set distance interval of 5 microns (minimum: 5 microns; maximum: 70 microns). Values are the *p-values* for the comparison of dendritic branching points between two groups at each distance from the cell soma (microns). All comparisons beyond 40 microns from the cell soma were not significant (S14 Table). A two-way analysis of variance (ANOVA) with Tukey's post-hoc test was used for statistical analysis of the Sholl data, which is graphically summarized in Fig 7. Significant *p-values* (< 0.05) are in **bold**.

## A graded mouse model of NEC

DSS administration to adult mice primarily leads to colitis [27], whereas in neonatal mice and rats, it causes small and large bowel inflammation [15,34]. We supplemented enteral feeds with different concentrations of DSS – 0.25%, 1%, 2% and 3% DSS – to create a graded model of NEC. To avoid maternal separation as a confounding factor [17], mice fed formula (0% DSS) served as controls for our experiments.

Although Ginzel et al. used 3% DSS supplementation to induce NEC in neonatal mice [15], we found that 3% DSS caused significant mortality within 48 hours (Fig 1A). The discrepancy in animal mortality and colitis severity may be due to differences in the mouse sub-strain or their microbiomes, which have been shown to vary in the same mouse strains from different laboratories [35]. Therefore, we used the 2% DSS concentration group to represent severe enterocolitis because 3% DSS-supplemented mice died too soon for adequate experimental analysis.

With increasing DSS supplementation, mortality increased (Fig 1A). Similarly, patients with more severe NEC (i.e., Bell's class III or surgical NEC) have a higher mortality rate [4]. Fullerton et al. showed that 74% of medically managed NEC patients survived to follow-up at 1–2 years of age, whereas only 62% of surgical NEC patients did [29]. Mortality from NEC animal protocols ranges from 0 to 30% [33]. In our experiments, formula-fed mice also had a significant mortality rate, highlighting the difficulty of gavage feeding of neonatal mice (2 grams) who are also stressed by separation from their dams [17].

The weight and clinical sickness score mirror the mortality data, where enterocolitis severity correlated with DSS supplementation (Figs 1B and 1C). Mice receiving high DSS supplementation (2%) developed worse clinical sickness scores earlier than those exposed to lower concentrations. Hence, we could prospectively control and create a graded NEC model by modulating enteral DSS supplementation.

Consistent with other groups [15,34], DSS exposure in neonatal mice resulted in enterocolitis and, to a lesser extent, colitis (Figs 2A-C), which contrasts with DSS's effects in adult mice [27]. Increased DSS supplementation resulted in worse macroscopic gut assessment scores (bowel distension, bloody stool, and friability), which correlates with NEC severity in other models [18]. Although there were no differences in apoptosis as indicated by cleaved caspase-3 staining, increased DSS supplementation decreased intestinal cell proliferation in a concentration-dependent manner, as shown by Ki-67 staining (Fig 2D). Our institution's IACUC protocol mandates euthanasia in moribund animals or animals with gross bloody stool. This may have accounted for the absence of caspase staining in our samples and decreased colitis, as animals may have been euthanized before developing fulminant colitis. Histological hallmarks of NEC such as villous sloughing and atrophy, necrosis and inflammatory infiltrates were seen on hematoxylin and eosin staining, all consistent with other models of NEC [21,33,36].

A critical component of NEC pathogenesis is the initiation and propagation of the systemic inflammatory response [28,37]. NEC patients have increased levels of cytokines and chemokines in the plasma (IL-1β, IL-6, IL-10, IL-17, TNF-α, CXCL1, CCL2, CCL3, and CCL4) [38–41]. Our study showed that all tested liver cytokines and chemokines were increased with higher DSS concentration, thus worsening NEC. In the plasma samples, several cytokines evaluated were below the detection limit, possibly due to cytokine and chemokine concentration fluctuations during the inflammatory process. IL-6, TNF-α, IFN-γ, IL-1β, and CXCL1 have been elevated in the plasma and in the brain in other animal models of NEC [12–15,34,40]. Our study looked at the most extensive panel of cytokines and chemokines to date.

## Implications for the gut-brain axis

Neurodevelopmental delay and sensorimotor disabilities are a major long-term morbidity found in NEC patients [42,43]. Severe NEC, especially if surgery is required, significantly increases the risk of cognitive and sensorimotor disabilities and affects patients' school performance [8]. Interestingly, preterm infants with NEC also have an increased risk for neurodevelopmental delays when compared to gestational age-matched controls without NEC history [6]. These results suggest that even mild NEC has the potential of affecting brain development; our mouse model has the potential to study mild

and severe NEC. Our findings that mice pups with mild degrees of enterocolitis, elicited through low concentration DSS supplementation to their feeds, show histological CNS changes with diminished branching complexity of microglia is consistent with these clinical reports.

Systemic cytokines and chemokines induced by NEC are believed to be one of the pathways whereby gut inflammation induces changes within the central nervous system during neurodevelopment [1]. NEC-induced inflammatory factors are hypothesized to enter the brain either directly across an immature blood-brain barrier or by causing the breakdown of the blood-brain barrier and allowing neuroinflammatory pathways and effector cells to be activated. We found that brain concentrations of IL-2, G-CSF, and CXCL1 increased with higher DSS concentrations (Fig 6, Table 2). There are many explanations for the increase of a smaller number of cytokines and chemokines in the brain during NEC. Pro- and anti-inflammatory cytokines have early and late-phase components; early-phase cytokines such as TNF-α may already be declining when measured in the brain in our NEC model. Alternately, it is another possibility that the blood-brain barrier may filter circulating cytokines and chemokines and thus limit the number of cytokines activating neuroinflammation [28]. The brain is an heterogenous organ so that specific regions may have cytokine and chemokine levels below the limit of detection by our methods. Both IL-2 and G-CSF knockout mice have neurodevelopmental deficits, suggesting that cytokines are critical for brain development in addition to their roles in the immune response [44,45]. Therefore, changes in cytokine expression may affect the balance of chemicals required for normal development in an immature central nervous system.

Microglia are a major mediator of neuroinflammation during NEC. Nino et al. have shown that reducing microglia activation by interfering with neuroinflammation pathways decreases white matter lesions and learning deficits in mice who have experienced NEC [11]. Although we did not observe an increase in microglia that has been reported [12,46], we observed a significant change in microglia activation in all mice exposed to DSS (Fig 7, Table 3), which is similar to the result of Sun et al. [14]. Therefore, our NEC model caused neuroinflammation by increasing cytokine and chemokine levels and activating microglia.

We did not find a difference in NeuN-positive mature neurons in mice with enterocolitis versus controls (S8 Fig). Our microglia and neuron results contrast with other groups, which found that NEC induction decreased mature neurons and increased microglia numbers in the hippocampus, basal ganglia, and cerebral cortex of pigs and mice [12,13,34]. However, these studies used different ages, induction protocols, and employed different animal species, where cell development and number may innately differ. The mouse age chosen in this model (postnatal days 3–6) mirrors the central nervous system development of third-trimester human infants, thus allowing us to study NEC's effect upon premature hosts. In contrast, many models examine this disease in mice whose neurodevelopmental ages are equivalent to term human infants [16].

In our graded NEC model, we can induce and control enterocolitis severity by varying DSS supplementation. Specific cytokine and chemokine levels in the liver, plasma, and brain correlate with the amount of DSS supplementation (Figs 3–6) and, thus, the severity of enterocolitis. Microglia demonstrate a more active morphological state in both mild (0.25% DSS) and severe NEC (Fig 7). Therefore, NEC affects brain cytokines, chemokines, and microglia; these factors, in turn, can interact and affect cell-cell cross-talk, circuit development, and, ultimately, normal brain development (Fig 8). Numerous cytokines have been shown to affect synaptic scaling, plasticity and circuit development [45,47,48]. Future research will focus on determining which inflammatory factors are responsible for widespread microglia activation, and the specific intracellular microglia pathways and functions that result in long-term neurological changes. This information can be used to identify targets and treatments to prevent or limit NEC-induced neurodevelopmental impairments.

### Limitations and considerations

Animal models are crucial for advancing our understanding of the gut-brain axis and for identifying mechanisms underlying the long-term neurodevelopmental deficits seen in NEC patients [13,18,21,33]. A major consideration in our study is the limited sample size. By testing several severities of NEC in neonatal mice and examining multiple tissues with different

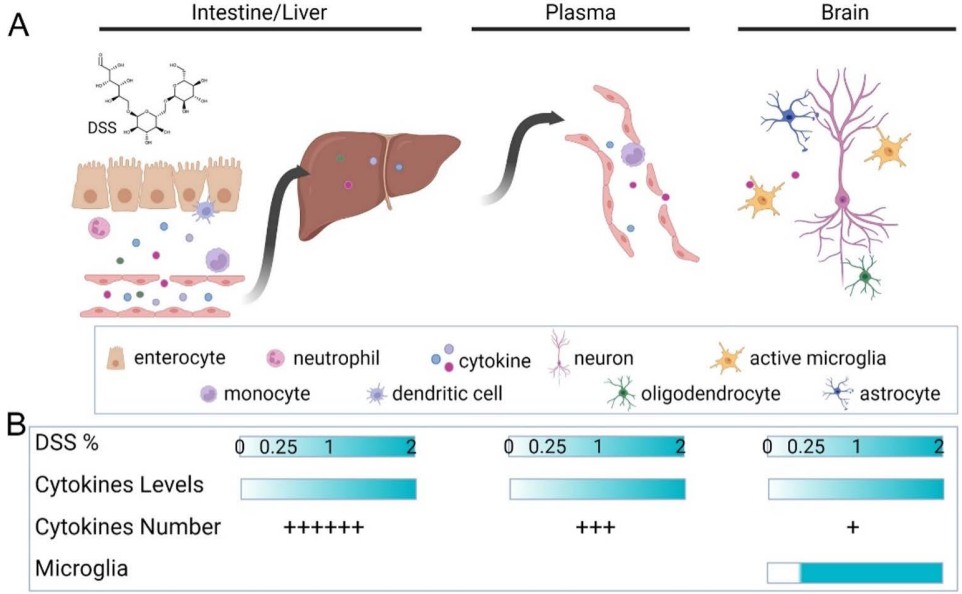

**Fig 8. Necrotizing Enterocolitis, cytokines, and neuroinflammation. (A)** DSS exposure induces intestinal damage and activation of innate immune responses in the neonatal gut with subsequent release of inflammatory cytokines and chemokines into the surrounding tissue. These inflammatory mediators reach the liver via the portal venous system. Systemic circulating inflammatory factors can reach the brain when they cross a leaky blood-brain barrier. In the brain, inflammatory cytokines and chemokines can interact with resident immune cells, microglia, or invading peripheral immune cells. These immune cells, cytokines, and chemokines can trigger neuroinflammation directly or indirectly on neurons, oligodendrocytes, and astrocytes. For example, cytokines may activate microglial engulfment of synaptic spines and affect circuit maturation during development. Activation of oligodendrocytes and astrocytes may affect myelination. **(B)** Inflammatory cytokine and chemokine expression increase with rising DSS concentrations and worsening enterocolitis. The number of cytokines whose expression correlates with DSS concentration decreases as we go from the liver to plasma and, eventually, to the brain. Compared to these cytokine patterns, microglia activation occurs even at low quantities of DSS supplementation and mild enterocolitis. Created in BioRender.com. Hsieh, H. (2025) https://BioRender.com/5hy3eg0.

experimental modalities, the number of samples were limited. Additionally, these results represent acute effects of NEC; future studies should include course experiments whereby one can study the evolution of mild NEC to severe NEC. Also, the results are not differentiated by sex, however, this will be an important factor for future studies. Finally, neurodevelopment in adult and adolescent animals that survived neonatal NEC can be evaluated with learning and memory paradigms and neuroanatomy.

The unique characteristics and strengths of our DSS rodent NEC model are 1) using a single hyperosmolar agent, DSS, to control NEC severity, 2) the mouse age, and 3) use of formula alone as a control. Despite differences in NEC induction technique, we see similar histopathological alterations and cytokine and chemokine changes than those observed in other animal models and in human NEC patients. We can induce and control NEC severity and inflammation, as evidenced by bowel histology, behavior, and cytokine and chemokine levels. Many mouse NEC mouse models use older mice (P9-11), whereas neurodevelopment in neonatal mice (P3-P6) in our study parallels human brain development during the third trimester [16]. The unique features of our model therefore allow studying the neurodevelopmental implications of NEC in the immature host. Maternal separation affects neurodevelopment; with the use of formula without DSS, we can control for these effects upon our animals. Although formula use can increase the risk for NEC, mouse breastmilk is not available to use.

## Summary

In conclusion, our graded neonatal mouse model of NEC enables researchers to study a wide range of NEC stages, including mild NEC. We have shown that NEC severity correlates with liver, plasma, and brain cytokines, which provide

another link to the gut-brain axis. This model will facilitate the identification of pathways involved in the progression of NEC, systemic inflammation, and sepsis, as well as cerebral inflammation and disruption of normal neurodevelopment. Ultimately, findings arising from this model may contribute towards identifying druggable targets for preventive and therapeutic strategies for different severities of NEC.

## Supporting information

**S1 Fig. Nursing mice exhibit notable differences in weight gain and cytokine profile compared to mice fed formula only (0% DSS).** Nursing mice gained weight more rapidly compared to formula-fed mice. Nursing mice nearly doubled their weight within 72 hours; two-way ANOVA with Tukey's post-hoc. Data presented as mean ± standard error of mean (SEM). ****$p < 0.0001$. Number of mice: nursing, 7; 0% DSS, 40. Because of these differences between nursing and 0% DSS-fed mice, as well as previous findings demonstrating that maternal separation in mice during the first week of life negatively affects neurodevelopment [17], we determined that isolating the mice during feeding is a confounding factor. Therefore, we used 0% DSS fed mice as controls for our experiments using different supplementations of DSS.
(PDF)

**S2 Fig. CSS, in each category, show the same overall trend during feeding as the combined CSS (relates to** Fig 1C**).** CSS measures were determined using a scoring system from Zani et al. (2008) (18). Data presented for each score component and stratified by feeding condition. (A) Appearance, (B) Natural activity, and (C) Response to touch scores increase over the course of the feeding protocol, even when animals are fed formula alone. Furthermore, higher concentrations of DSS are associated with worse CSS earlier. Two-way ANOVA with Tukey's post-hoc, $p < 0.0001$ for all analyses. Data presented as mean ± SEM. Number of mice: 0%, 29; 0.25%, 26; 1%, 26; 2%, 7.
(PDF)

**S3 Fig. External bowel scores, by category, show the same overall trend as the combined external bowel score (relates to** Fig 2B**).** Intestines were assessed as described in Zani et al (2008) [18]. Bowel (A) consistency, (B) color, and (C) dilation are significantly more severe in mice fed with increasing concentrations of DSS (1% and 2% DSS), compared to lower concentrations of DSS (0% and 0.25% DSS). One-way ANOVA with Tukey's post-hoc, $p < 0.0001$ for all analyses. Data presented as mean ± SEM. ***$p < 0.001$, ****$p < 0.0001$. Number of mice: 0%, 18; 0.25%, 13; 1%, 18; 2%, 7.
(PDF)

**S4 Fig. Cleaved caspase-3 does not appear in the small intestinal crypts across all experimental groups.** Representative images of the small intestine imaged at 10x then magnified an additional 20x (scale 100 μm). Brown staining represents cleaved capase-3 (CC3), a marker for cellular apoptosis, which is not present in any of the small intestinal crypts. Positive CC3 staining on the tips of the small intestinal villi is not quantifiable because normal conditions also have cellular apoptosis at this location. Number of mice: 0%, 4; 0.25%, 6; 1%, 4; 2%, 4.
(PDF)

**S5 Fig. Some blood plasma cytokine and chemokine concentrations do not significantly correlate with DSS concentration (relates to** Fig 5**).** Cytokines/chemokines include (A) IL-1α, $p = 0.83$, (B) IL-6, $p = 0.070$, and (C) CCL3, $p = 0.054$. Simple linear regression with log-transformation of y values was performed. Data presented as boxplots showing min-max. Slope and y intercept with confidence intervals are plotted. $ns$ = not significant ($p \geq 0.05$). Number of mice: 0%, 14; 0.25%, 11; 1%, 14; 2%, 4.
(PDF)

**S6 Fig. Many brain cytokine concentrations do not significantly correlate with DSS concentration (relates to** Fig 6**).** In contrast, other cytokines tested in the brain did not show the same trend. These cytokines include **(A)**

IL-1α, $p = 0.074$, **(B)** IL-1β, $p = 0.25$, **(C)** IL-6, $p = 0.47$, **(D)** IL-10, $p = 0.14$, **(E)** IL-12(p40), $p = 0.98$, **(F)** IL-12(p70), $p = 0.75$, **(G)** IL-17, $p = 0.096$, **(H)** TNF-α, $p = 0.94$, **(I)** IFN-γ, $p = 0.11$, and **(J)** GM-CSF, $p = 0.089$. Simple linear regression with log-transformation of y values was performed. Data presented as boxplots showing min-max. Slope and y intercept with confidence intervals are plotted. *ns* = not significant ($p \geq 0.05$). Number of mice: 0%, 12; 0.25%, 6; 1%, 10; 2%, 3.
(PDF)

**S7 Fig. Many brain chemokine concentrations do not significantly correlate with DSS concentration (relates to** Fig 6**).** Other chemokines do not show significant trends in the brain; these chemokines include (A) CCL2, $p = 0.29$, (B) CCL3, $p = 0.13$, and (C) CCL4, $p = 0.46$. Simple linear regression with log-transformation of y values was performed. Data presented as boxplots showing min-max. Slope and y intercept with confidence intervals are plotted. *ns* = not significant ($p \geq 0.05$). Number of mice: 0%, 12; 0.25%, 6; 1%, 10; 2%, 3.
(PDF)

**S8 Fig. No significant differences of neuron or microglia proportions in CA1 hippocampal region across all DSS experimental groups.** (A) Representative immunohistochemical images (30x magnification) of the CA1 hippocampal region. Slices are stained blue for DAPI, red for neurons, and green for microglia. Scale bar = 50μm. (B-C) The proportion of (B) neurons, $p = 0.046$, and (C) microglia, $p = 0.018$, out of all cells in the CA1 region is comparable across all DSS concentrations. A significant difference was only found when comparing the proportions of neurons between mice fed 1% and 2% DSS, $p = 0.027$ (S12 Table), and when comparing the proportions of microglia between mice fed 0.25% and 1% DSS, $p = 0.018$ (S13 Table). One-way ANOVA with Tukey's post-hoc. Data presented as mean ± SEM. *$p < 0.05$. n = 9 immunohistochemical images for all experimental groups.
(PDF)

**S1 Table. Comparisons of Kaplan-Meier survival curves (relates to** Fig 1A**).** *P-values* for the comparison between the survival curves of two experimental groups (indicated in the first row and column). A Kaplan-Meier survival analysis was used for mortality during feeding. Significant *p-values* (< 0.05) are in **bold**.
(PDF)

**S2 Table. Normalized weights of mice that survived to the final 72-hour timepoint (relates to** Fig 1B**).** Mean weights, standard error of mean (SEM), and counts of all mice that survived the entire feeding protocol.
(PDF)

**S3 Table. Comparisons of normalized weight curves during feeding (relates to** Fig 1B**).** *P-values* for the comparison between weight gain curves of two experimental groups (indicated in the first row and column). A two-way analysis of variance (ANOVA) with Tukey's post-hoc was used for statistical analysis of weight gain curves. Significant *p-values* (< 0.05) are in **bold**.
(PDF)

**S4 Table. Clinical sickness scores (CSS) of mice during feeding (relates to** Fig 1C**).** Mean ± SEM for the behavior score at each feeding time (in hours); the values in the row below are the number of mice alive during that feeding time for behavioral assessment. CSS measures were determined using a scoring system from Zani et al. (2008) [18].
(PDF)

**S5 Table. Comparisons of CSS during feeding (relates to** Fig 1C**).** *P-values* for the comparison of behavioral scores between two groups at each feeding time (in hours). A two-way ANOVA with Tukey's post-hoc test was used for statistical analysis of the behavioral scores during feeding. Significant *p-values* (< 0.05) are in **bold**.
(PDF)

**S6 Table. External bowel scores of control and DSS-fed mice (relates to Fig 2B).** Mean external bowel scores, SEM, and counts of mice whose intestines were evaluated macroscopically. Bowels were evaluated using a scoring system from Zani et al. (2008) [18].
(PDF)

**S7 Table. Comparisons of external bowel scores (relates to Fig 2B)** . *P-values* for the comparison of external bowel scores between two groups (indicated in the first row and column). A one-way ANOVA with Tukey's post-hoc test was used for statistical analysis of the external bowel scores. Significant *p-values* (< 0.05) are emphasized in **bold**.
(PDF)

**S8 Table. Comparisons of intestinal pathology scores across bowel regions (relates to Fig 2C).** *P-values* for the comparison of intestinal pathology scores between two groups at each region of the GI tract. A two-way ANOVA with Tukey's post-hoc test was used for statistical analysis of the intestinal pathology scores. Significant *p-values* (< 0.05) are in ***bold***.
(PDF)

**S9 Table. Comparisons of intestinal pathology scores across DSS concentrations (relates to Fig 2C).** *P-values* for the comparison of intestinal pathology scores between two groups at each region of the GI tract. A two-way ANOVA with Tukey's post-hoc test was used for statistical analysis of the intestinal pathology scores. Significant *p-values* (< 0.05) are in ***bold***.
(PDF)

**S10 Table. Ki-67+ cell counts per small intestinal crypt in mice (relates to Fig 2D).** Mean Ki-67+ cell count, SEM, and number of all intestinal crypts counted.
(PDF)

**S11 Table. Comparisons of Ki-67+ scores (relates to Fig 2D).** *P-values* for the comparison of Ki-67+ small intestinal crypt counts between two groups (indicated in the first row and column). A one-way ANOVA with Tukey's post-hoc test was used for statistical analysis of the Ki-67+ counts. Significant *p-values* (< 0.05) are in ***bold***.
(PDF)

**S12 Table. Comparisons of neuron proportions in CA1 hippocampus (relates to S8B Fig).** *P-values* for the comparison between two groups (indicated in the first row and column) of the proportion of neurons among all cells in the CA1 hippocampal region. A one-way ANOVA with Tukey's post-hoc test was used for statistical analysis of the neuron proportions. Significant *p-values* (< 0.05) are in ***bold***.
(PDF)

**S13 Table. Comparisons of microglia proportions in CA1 hippocampus (relates to S8C Fig).** *P-values* for the comparison between two groups (indicated in the first row and column) of the proportion of microglia among all cells in the CA1 hippocampal region. A one-way ANOVA with Tukey's post-hoc test was used for statistical analysis of the microglia proportions. Significant *p-values* (< 0.05) are in ***bold***.
(PDF)

**S14 Table. Sholl analysis comparisons at distances beyond 40 microns from microglial cell soma (relates to Fig 7 and Table 3).** Sholl analysis was used to quantify dendritic branching complexity of microglia, with a set distance interval of 5 microns (minimum: 5 microns; maximum: 70 microns). Values are the *p-values* for the comparison of dendritic branching points between two groups at each distance from the cell soma (microns). A two-way analysis of variance (ANOVA) with Tukey's post-hoc test was used for statistical analysis of the Sholl data, which is graphically summarized in Fig 7. Significant *p-values* (< 0.05) are in ***bold***.
(PDF)

## Acknowledgments

The authors would like to thank Drs. Gail Besner, Amalia Napoli, Vincent Yang, Ashwini Phadnis-Moghe, Miguel Maderia, and Yijie Wang for scientific discussion and assistance with this manuscript. The authors would also like to thank Dr. Amrendra Singh, Noah Teaney, Sarah Bucher, and Stella Ku for their contributions to data collection.

## Author contributions

**Conceptualization:** Lonnie P Wollmuth, Esther M Speer, Helen Hsieh.

**Data curation:** Cuilee Sha, Trevor Van Brunt, Jacob Kudria, Donna Schmidt, Jela Bandovic, Michael Giarrizzo, Esther M Speer, Helen Hsieh.

**Formal analysis:** Cuilee Sha, Trevor Van Brunt, Jacob Kudria, Alisa Yurovsky, Joyce Lin, Agnieszka B. Bialkowska, Esther M Speer, Helen Hsieh.

**Funding acquisition:** Agnieszka B. Bialkowska, Lonnie P Wollmuth, Esther M Speer, Helen Hsieh.

**Investigation:** Cuilee Sha, Trevor Van Brunt, Jela Bandovic, Agnieszka B. Bialkowska, Lonnie P Wollmuth, Esther M Speer, Helen Hsieh.

**Methodology:** Cuilee Sha, Trevor Van Brunt, Jacob Kudria, Donna Schmidt, Styliani-Anna Tsirka, Helen Hsieh.

**Project administration:** Lonnie P Wollmuth, Helen Hsieh.

**Resources:** Cuilee Sha, Helen Hsieh.

**Software:** Cuilee Sha, Jacob Kudria, Helen Hsieh.

**Supervision:** Lonnie P Wollmuth, Esther M Speer, Helen Hsieh.

**Validation:** Cuilee Sha, Helen Hsieh.

**Visualization:** Cuilee Sha, Helen Hsieh.

**Writing – original draft:** Cuilee Sha, Trevor Van Brunt, Jacob Kudria, Agnieszka B. Bialkowska, Lonnie P Wollmuth, Helen Hsieh.

**Writing – review & editing:** Cuilee Sha, Styliani-Anna Tsirka, Agnieszka B. Bialkowska, Lonnie P Wollmuth, Esther M Speer, Helen Hsieh.

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
