## [Decision Letter · Decision Letter 0]

10 Mar 2025

PONE-D-25-05095A graded neonatal mouse model of necrotizing enterocolitis demonstrates that mild enterocolitis is sufficient to activate microglia and increase cerebral cytokine expressionPLOS ONE

Dear Dr. Hsieh,

Thank you for submitting your manuscript to PLOS ONE. After careful consideration, we feel that it has merit but does not fully meet PLOS ONE’s publication criteria as it currently stands. Therefore, we invite you to submit a revised version of the manuscript that addresses the points raised during the review process.

We look forward to receiving your revised manuscript.

Kind regards,

Kazumichi Fujioka

Academic Editor

PLOS ONE

Journal Requirements:

This work was supported by the following grants: Targeted Research Opportunity Grant from the Office of the Vice President of Research at Stony Brook University (ES, HH), NIH Grants R01 DK124342 (ABB), R01 NS088479 (LPW).

Reviewers' comments:

Reviewer's Responses to Questions

**Comments to the Author**

1. Is the manuscript technically sound, and do the data support the conclusions?

Reviewer #1: Yes

Reviewer #2: Yes

2. Has the statistical analysis been performed appropriately and rigorously? 

Reviewer #1: Yes

Reviewer #2: Yes

3. Have the authors made all data underlying the findings in their manuscript fully available?

Reviewer #1: Yes

Reviewer #2: Yes

4. Is the manuscript presented in an intelligible fashion and written in standard English?

Reviewer #1: Yes

Reviewer #2: Yes

5. Review Comments to the Author

Reviewer #1: In the study, the relationship between NEC severity and neuroinflammation was investigated with the new NEC model.

That's why I think the article is important. A well-planned study in terms of hypothesis and methodology. I think it is suitable for publication

Reviewer #2: First of all, I would like to congratulate the authors for the research work they have carried out. This is a necessary research objective at this time due to the lack of evidence in this regard, especially at the level of seeking therapeutic strategies against the sometimes devastating neurological effects of necrotizing enterocolitis. I only ask the authors a few brief questions and/or recommendations.

1) I would like to understand why you have selected only those two antibodies in the immunohistochemistry? Have you considered MBP, NFH or SMI32? Why have you focused on the hippocampus?

2) Were the brain sections coronal or sagittal? Please clarify in the text.

3) In the discussion when you talk about microglia activation, I would encourage you to make a proposal on where future research should go, especially focused on limiting neurological injury in murine models.

Thank you for your excellent work.

6. PLOS authors have the option to publish the peer review history of their article (what does this mean? ). If published, this will include your full peer review and any attached files.

**Do you want your identity to be public for this peer review?** For information about this choice, including consent withdrawal, please see our Privacy Policy .

Reviewer #1: No

Reviewer #2: **Yes: ** Felipe Garrido PhD MD

---

## [Author Response · Author response to Decision Letter 1]

1 Apr 2025

Response to Reviewers. PLOS ONE.

Manuscript# PONE-D-25-05095. “A graded neonatal mouse model of necrotizing enterocolitis demonstrates that mild enterocolitis is sufficient to activate microglia and increase cerebral cytokine expression.”

We greatly appreciate the Editor and Reviewers for their thoughtful comments. As outlined below, we have incorporated all comments/suggestions/requests into the revised version of the manuscript in some form. Overall, the comments, critiques, and suggestions by the Editor/Reviewers have greatly improved the clarity and rigor of the manuscript.

Original comments by Editor or Reviewers in italics. We present our point-by-point responses to comments/requests below. Also note that indicated page numbers are for the “Revised Manuscript without Track Changes” version of the manuscript.

Requests from Editor & Journal Requirements:

The Editor noted several concerns from the Reviewers. We give more detailed responses below in response to comments from the specific Reviewer.

We appreciate the inclusion of the journal’s style templates, and we apologize for not adhering to them more closely in our original submission. We have now completely reformatted our manuscript to match what is exemplified in both style templates. If there is anything we have missed, we are happy to correct it in a timely manner.

Additionally, we have uploaded all figure files into PLOS’s Preflight Analysis and Conversion Engine (PACE) digital diagnostic tool, https://pacev2.apexcovantage.com/ to ensure that they meet PLOS ONE’s file requirements.

We apologize for the mismatch between our Funding Information and Financial Disclosure sections. We have now edited the “Funding Information” section to include the correct information, which we have rechecked amongst all our collaborators.

This work was supported by the following grants: Targeted Research Opportunity Grant from the Office of the Vice President of Research at Stony Brook University (ES, HH), NIH Grants R01 DK124342 (ABB), R01 NS088479 (LPW).

As requested, we will include the suggested phrase and have updated the funding support for the authors. Therefore, our financial disclosure statement is:

“This work was supported by the following grants: Targeted Research Opportunity Grant from the Office of the Vice President of Research at Stony Brook University (ES, HH), NIH Grants R01 DK124342 (ABB), R01 DK052230 (VWY), R01 NS088479 (LPW). The funders had no role in study design, data collection and analysis, decision to publish, or preparation of the manuscript.”

We noted three instances in our manuscript where the phrase “data not shown” or similar appears and have addressed them accordingly:

(1) Fig 1 caption read “3% DSS data not shown because mice were too sick to be assessed”. We have changed this to “3% DSS data not collected because mice were too sick to be assessed” to more accurately reflect our experimental procedure.

(2) Section: Results, Subsection: Increased DSS exposure suppresses intestinal cell proliferation without promoting apoptosis

We had originally written “…small intestinal crypts did not show any positive staining for CC3 (data not shown).” We have revised this to include S4 Fig, which contains representative images of CC3 immunohistochemistry in the small intestines to demonstrate the lack of staining within the small intestinal crypts amongst all experimental groups. This supports our conclusion that apoptosis does not appear to play a role in intestinal disarray resulting from DSS administration.

(3) Table 3 caption read “All comparisons beyond 40 microns from the cell soma were not significant (data not shown)”. We have revised this to include S14 Table (line 494, pp. 14), which contains all the comparisons among microglia branching morphologies beyond 40 microns.

We have now checked each reference against both PubMed and Retraction Watch Database, which can be found at https://retractiondatabase.org/RetractionSearch.aspx. We found no references that were retracted or changed since their initial publication.

We did, however, find that we had accidentally duplicated one of our references; reference 12 and 47 refer to the same publication. This error is now corrected so that reference 47 is removed and the in-text citation has been replaced with reference 12.

If the journal has identified any retracted publications that we have missed, please let us know, and we are happy to correct it in a timely manner.

Reviewer #1 (Comments to the Authors):

In the study, the relationship between NEC severity and neuroinflammation was investigated with the new NEC model.

That's why I think the article is important. A well-planned study in terms of hypothesis and methodology. I think it is suitable for publication.

We greatly appreciate the Reviewer’s kind words, especially regarding the significance and study design of our manuscript.

Reviewer #2 (Comments to the Authors):

First of all, I would like to congratulate the authors for the research work they have carried out. This is a necessary research objective at this time due to the lack of evidence in this regard, especially at the level of seeking therapeutic strategies against the sometimes devastating neurological effects of necrotizing enterocolitis. I only ask the authors a few brief questions and/or recommendations.

We greatly appreciate the Reviewer’s kind words.

1) I would like to understand why you have selected only those two antibodies in the immunohistochemistry? Have you considered MBP, NFH or SMI32? Why have you focused on the hippocampus?

The Reviewer raises numerous interesting points with these questions, which we address below:

(1) We were specifically interested in seeing whether NEC induction would result in an acute neuroinflammatory response. Using Iba1, we could directly assay microglia numbers and activation status using morphology as proxy. Microglia are activated by NEC and have been shown to be important for the pathogenesis of NEC neuroinflammation (Nino et al, 2019). Infants with NEC demonstrate decreased brain volumes (both gray and white matter), therefore, for this study, we used NeuN as a marker for mature neurons. Several other animal models showed changes in neuronal number, and we wanted to see if it is seen in our model (Biouss et al., 2019, Sun et al., 2018).

(2) We greatly appreciate the suggestion to consider these markers by the Reviewer. We believe that these makers of myelin (MBP) and neurodegeneration (NFH and SMI32) are better suited for more long-term studies of animals after they grow into adolescence following NEC induction. Although MBP expression begins at birth, its production peaks a little later (Cristobal and Lee, 2022). For this manuscript, we first wanted to focus on acute inflammatory effects.

We are looking at MBP in our long-term studies. For the NFH and SMI32 specifically, we did not observe a change in our neuronal staining with NeuN, therefore, we had not looked at these markers. Nevertheless, it would be interesting to see if there is a change in neuronal morphology with NFH and SMI32. We will apply it to future studies.

(3) The hippocampus is a well-studied region of the brain that is associated with learning and memory. These specific tasks are often deficient in adolescent patients who have been diagnosed with NEC neonatally. Biouss et al. (2019) demonstrated decreases in number of neurons, oligodendrocytes and neuronal precursors in the hippocampus and not in the cortex. Sun et al (2018) find increased amoeboid phenotype in hippocampal microglia in a porcine model of NEC.

2) Were the brain sections coronal or sagittal? Please clarify in the text.

Thank you for requesting clarification. The brain sections were all coronal. This is now included in the revised manuscript in the Materials and Methods section.

3) In the discussion when you talk about microglia activation, I would encourage you to make a proposal on where future research should go, especially focused on limiting neurological injury in murine models.

We thank the Reviewer for this feedback and recommendation. We revised the manuscript on line 596-599 (pp. 16) to expand further on potential future research given our findings on microglia activation and cytokine/chemokine correlations with varying severities of NEC.

Thank you for your excellent work.

We would like to once again thank the Reviewer, and we appreciate the Reviewer’s comments and insightful suggestions.

---

## [Decision Letter · Decision Letter 1]

11 Apr 2025

A graded neonatal mouse model of necrotizing enterocolitis demonstrates that mild enterocolitis is sufficient to activate microglia and increase cerebral cytokine expression

PONE-D-25-05095R1

Dear Dr. Hsieh,

We’re pleased to inform you that your manuscript has been judged scientifically suitable for publication and will be formally accepted for publication once it meets all outstanding technical requirements.

Kind regards,

Kazumichi Fujioka

Academic Editor

PLOS ONE

Additional Editor Comments (optional):

Reviewers' comments:

Reviewer's Responses to Questions

**Comments to the Author**

1. If the authors have adequately addressed your comments raised in a previous round of review and you feel that this manuscript is now acceptable for publication, you may indicate that here to bypass the “Comments to the Author” section, enter your conflict of interest statement in the “Confidential to Editor” section, and submit your "Accept" recommendation.

Reviewer #2: All comments have been addressed

2. Is the manuscript technically sound, and do the data support the conclusions?

Reviewer #2: Yes

3. Has the statistical analysis been performed appropriately and rigorously? 

Reviewer #2: Yes

4. Have the authors made all data underlying the findings in their manuscript fully available?

Reviewer #2: Yes

5. Is the manuscript presented in an intelligible fashion and written in standard English?

Reviewer #2: Yes

6. Review Comments to the Author

Reviewer #2: (No Response)

7. PLOS authors have the option to publish the peer review history of their article (what does this mean? ). If published, this will include your full peer review and any attached files.

**Do you want your identity to be public for this peer review?** For information about this choice, including consent withdrawal, please see our Privacy Policy .

Reviewer #2: No

---

## [Editor Report · Acceptance letter]

PONE-D-25-05095R1

PLOS ONE

Dear Dr. Hsieh,

I'm pleased to inform you that your manuscript has been deemed suitable for publication in PLOS ONE. Congratulations! Your manuscript is now being handed over to our production team.

Kind regards,

on behalf of

Dr. Kazumichi Fujioka

Academic Editor

PLOS ONE